# Neurology-related protein biomarkers are associated with cognitive ability and brain volume in older age

Sarah E. Harris [1,2]*, Simon R. Cox [1,2,3], Steven Bell [4,5,6], Riccardo E. Marioni[1,7], Bram P. Prins[4],
Alison Pattie[2], Janie Corley[1,2], Susana Muñoz Maniega[1,8,9], Maria Valdés Hernández[1,8,9], Zoe Morris[8],
Sally John[10], Paola G. Bronson[10], Elliot M. Tucker-Drob[11], John M. Starr[1,12], Mark E. Bastin[1,3,8],
Joanna M. Wardlaw [1,3,8,9], Adam S. Butterworth[4,5] & Ian J. Deary[1,2]

Identifying biological correlates of late life cognitive function is important if we are to ascertain biomarkers for, and develop treatments to help reduce, age-related cognitive decline. Here, we investigated the associations between plasma levels of 90 neurology-related proteins (Olink® Proteomics) and general fluid cognitive ability in the Lothian Birth Cohort 1936 (LBC1936, N = 798), Lothian Birth Cohort 1921 (LBC1921, N = 165), and the INTERVAL BioResource (N = 4451). In the LBC1936, 22 of the proteins were significantly associated with general fluid cognitive ability (β between −0.11 and −0.17). MRI-assessed total brain volume partially mediated the association between 10 of these proteins and general fluid cognitive ability. In an age-matched subsample of INTERVAL, effect sizes for the 22 proteins, although smaller, were all in the same direction as in LBC1936. Plasma levels of a number of neurology-related proteins are associated with general fluid cognitive ability in later life, mediated by brain volume in some cases.

[1] Centre for Cognitive Ageing and Cognitive Epidemiology, University of Edinburgh, 7 George Square, Edinburgh EH8 9JZ, UK. [2] Department of Psychology, University of Edinburgh, 7 George Square, Edinburgh EH8 9JZ, UK. [3] Scottish Imaging Network, A Platform for Scientific Excellence (SINAPSE) Collaboration, 300 Bath St, Glasgow, UK. [4] UK Medical Research Council/British Heart Foundation Cardiovascular Epidemiology Unit, Department of Public Health and Primary Care, University of Cambridge, Strangeways Research Laboratory, Wort's Causeway, Cambridge CB1 8RN, UK. [5] The National Institute for Health Research Blood and Transplant Unit in Donor Health and Genomics at the University of Cambridge, University of Cambridge, Strangeways Research Laboratory, Wort's Causeway, Cambridge CB1 8RN, UK. [6] Stroke Research Group, Department of Clinical Neurosciences, University of Cambridge Neurology Unit, Cambridge Biomedical Campus, Cambridge CB20QQ, UK. [7] Centre for Genomic and Experimental Medicine, MRC Institute of Genetics and Molecular Medicine, The University of Edinburgh, Western General Hospital, Crewe Road, Edinburgh EH4 2XU, UK. [8] Brain Research Imaging Centre, Neuroimaging Sciences, The University of Edinburgh, Chancellor's Building, 49 Little France Crescent, Edinburgh, UK. [9] UK Dementia Research Institute at the University of Edinburgh, Edinburgh BioQuarter, Edinburgh, UK. [10] Translational Biology, Biogen, Cambridge, MA 02142, USA. [11] Department of Psychology, University of Texas, 108 E Dean Keeton St, Austin, TX, USA. [12] Alzheimer Scotland Dementia Research Centre, University of Edinburgh, 7 George Square, Edinburgh EH8 9JZ, UK. *email: Sarah.Harris@igmm.ed.ac.uk

As populations in developed countries continue to age, there is a growing need to understand the biological correlates of individual differences in cognitive ability in later life. Ageing-related cognitive changes are thought to be driven—at least in part—by structural changes in the brain[1]. For example, global atrophy, grey matter and white matter volumes, white matter microstructure and measures such as white matter hyperintensities (WMH) and perivascular spaces (PVS)—which are markers of cerebral small-vessel disease (SVD)—have been associated with reduced cognitive ability and risk of dementia in both cross-sectional and longitudinal studies[2–7].

Large-scale genome-wide association studies have shown that cognitive ability in later life is highly heritable and polygenic[8–12]. Due to the highly polygenic nature of this trait, it is challenging to identify relevant biological pathways from the genetic variants associated with it. However, gene expression is itself determined by a combination of genetic, ontogenetic and environmental factors. Because proteins are the proximal products of transcribed and expressed genetic code, directly measuring protein levels can increase power to identify biological pathways in later-life cognitive function. Protein levels are more directly linked than genetic variants to individual variation in cognitive function and structural brain phenotypes, with post-translational buffering as a potential mechanism for mitigating many environmental factors[13]. Peripheral blood proteins, including inflammatory markers[14,15] and S100β[16], have previously been associated with cognitive ability and/or MRI brain measures, but until recently, it has been relatively difficult and cost-prohibitive to measure multiple proteins in large numbers of plasma samples[17], which is what is required if we are to develop biomarkers of cognitive function in later life in an easily accessible biological sample. Technological advances have enabled high-throughput and cost-effective measurement of plasma proteins, enabling us to link plasma proteomics to cognitive function and brain structure in three large population samples for the first time.

In this study we measured 90 neurology-related protein biomarkers using the Proseek Multiplex Neurology I 96 × 96 reagents kit produced by Olink® Proteomics (Uppsala, Sweden) [18,19]. These proteins have been implicated in neurological processes and/or diseases, cellular regulation, immunology, development or metabolism[20]. The proteins were selected based on literature text mining and assay performance. The participants were ~800 members of the Lothian Birth Cohort 1936 (LBC1936)[21], ~170 members of the older Lothian Birth Cohort 1921 (LBC1921)[21] and ~4500 members of INTERVAL, split into a LBC1936 age-matched subsample and a younger subsample to investigate if associations were consistent across different age groups[22]. In cross-sectional analyses we investigated the association of 90 plasma proteins with general fluid cognitive ability in 5414 samples. In the LBC1936 cohort we tested for association with brain volumes (total brain, grey matter and normal-appearing white matter, WMH), PVS and white matter tract measures derived from quantitative tractography (fractional anisotropy [FA], mean diffusivity [MD]). We investigated whether any associations between the neurology-related plasma protein levels and general fluid cognitive ability were mediated by structural brain variables. We hypothesised that some of the neurology-related proteins would be associated with general fluid cognitive ability in older individuals, and that some of these associations would be mediated by structural brain variables.

We identify 22 neurology-related proteins that are associated with general fluid cognitive ability in later life in the LBC1936, ten of which are mediated by total brain volume. Effect sizes for the 22 proteins, although smaller, are all in the same direction as in LBC1936 in an age-matched subsample of INTERVAL. Similar effect sizes are found for the majority of these 22 proteins in the older LBC1921. The associations are not replicated in a younger subset of INTERVAL. In conclusion, we identify plasma levels of a number of neurology-related proteins that are associated with general fluid cognitive ability in later life, some of which are mediated by brain volume.

## Results

**Descriptive statistics**. Descriptive statistics for general fluid cognitive ability in the LBC1936, LBC1921, INTERVAL-Old and INTERVAL-Young samples and for the brain magnetic resonance imaging (MRI) variables (LBC1936 only) are shown in Tables 1 and 2.

**PCA of the 90 neurology-related protein biomarkers**. Principal component analysis (PCA) indicated that, for all four cohorts, the majority of the variance in the protein data was explained by the first 17 components (63%–74%), with greater than 30% explained by principal component (PC) 1 (Supplementary Data 1, Fig. 1). The component loadings for PC1–PC5 are shown in Supplementary Tables 1–4. The coefficient of factor congruence between the four cohorts ranged between |0.85 and 1.00| for the first three principal components (Supplementary Data 1, Fig. 2). Therefore, protein–PC1–PC3 were selected for

### Table 1 Summary descriptive data for LBC1936.

| Variable | Mean (SD, range) | N (with OLINK data) |
|---|---|---|
| Age at cognitive testing and plasma collection (years) | 72.5 (0.7, 70.9–74.2) | 805 |
| General fluid cognitive ability | −0.0 (1.0, −3.5–3.2) | 798 |
| Age at brain scan (years) | 72.7 (0.7, 71.0–74.2) | 684 |
| Total brain volume ($cm^3$) | 989.1 (90.4, 730.8–1247.0) | 600 |
| Grey matter volume ($cm^3$) | 471.7 (45.1, 366.4–616.2) | 600 |
| Normal-appearing white matter volume ($cm^3$) | 474.6 (51.0, 301.7–636.0) | 600 |
| White matter hyperintensity volume ($cm^3$) | 12.1 (13.0, 0.0–98.4) | 617 |
| Intracranial volume ($cm^3$) | 1438.7 (135.3, 1059.3–1857.9) | 618 |
| General fractional anisotropy | 0.0 (0.02, −0.07–0.05) | 621 |
| General mean diffusivity | −0.0 (29.2, −76.6–92.7) | 621 |
| Sex | Male | 423 (52.5%) |
|  | Female | 382 (47.5%) |
| Smoking status | Smoker | 68 (8.5%) |
|  | Ex-smoker | 353 (43.9%) |
|  | Never smoker | 384 (47.7%) |
| Antihypertensive medication | Yes | 373 (46.3%) |
|  | No | 432 (53.7%) |

**Table 2 Summary descriptive data for INTERVAL-Old, INTERVAL-Young and LBC1921.**

| Variable | | INTERVAL-Old | | INTERVAL-Young | | LBC1921 | |
|---|---|---|---|---|---|---|---|
| | | Mean (SD, range) | N (with OLINK data) | Mean (SD, range) | N (with OLINK data) | Mean (SD, range) | N (with OLINK data) |
| Age (years) | | 70.3 (2.4, 67.0–77.8) | 975 | 58.4 (5.0, 48.9–66.9) | 3476 | 86.6 (0.4, 85.7–87.4) | 175 |
| General fluid cognitive ability | | −1.2 (1.4, −8.9–3.2) | 975 | −0.5 (1.3, −6.6–3.4) | 3476 | 0.0 (1.0, −2.6–3.4) | 165 |
| Sex | Male | 646 (66.3%) | | 1975 (56.8) | | 83 (47.4%) | |
| | Female | 329 (33.7%) | | 1501 (43.2%) | | 92 (52.6%) | |
| Smoking status | Smoker | 28 (2.9%) | | 162 (4.7%) | | NA | NA |
| | Ex-smoker | 493 (51.1%) | | 1284 (37.4%) | | NA | NA |
| | Never smoker | 444 (46.0%) | | 1986 (57.9%) | | NA | NA |
| Antihypertensive medication use | Yes | 166 (17.2%) | | 378 (11.0%) | | NA | NA |
| | No | 799 (82.80) | | 3054 (89.0%) | | NA | NA |
| Trial arm | F12 | 107 (11.1%) | | 480 (14.0%) | | NA | NA |
| | F14 | 98 (10.2%) | | 500 (14.6%) | | NA | NA |
| | F16 | 119 (12.3%) | | 507 (14.8%) | | NA | NA |
| | M08 | 210 (21.8%) | | 639 (18.6%) | | NA | NA |
| | M10 | 230 (23.8%) | | 677 (19.7%) | | NA | NA |
| | M12 | 201 (20.8%) | | 629 (18.3%) | | NA | NA |

F## and M## = female—inter-donation interval in weeks; male—inter-donation interval in weeks.

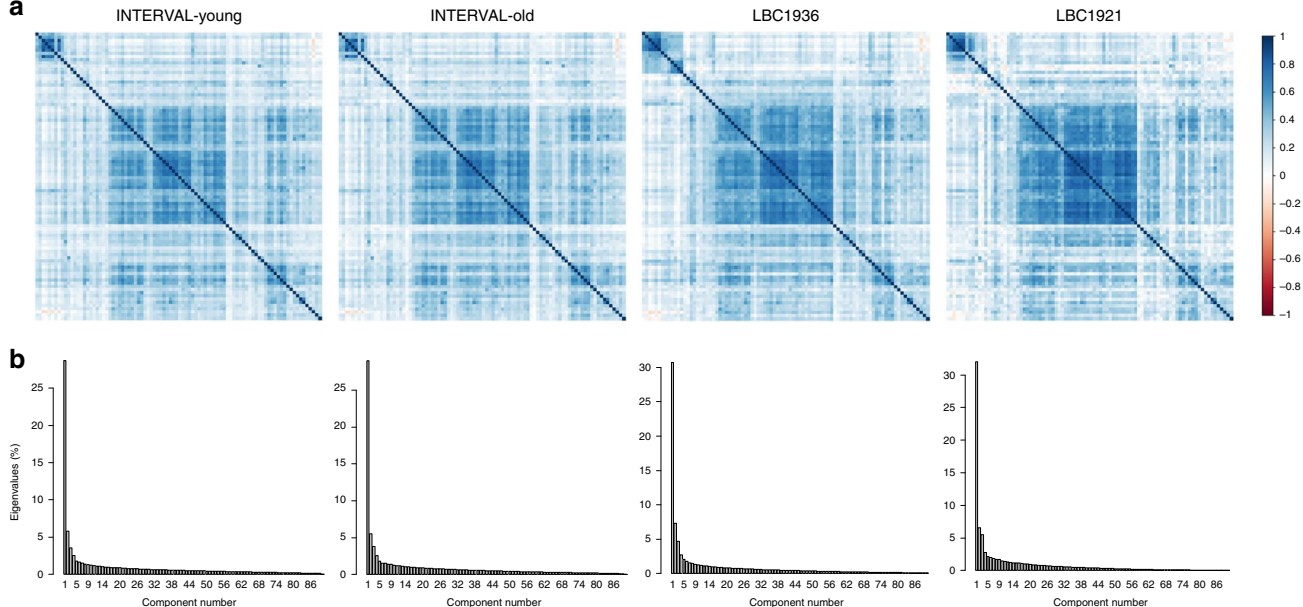

**Fig. 1 Principal component analysis on the 90 neurology-related proteins in INTERVAL-Young, INTERVAL-Old, LBC1936 and LBC1921. a** Heatmaps and **b** screenplots. Source data are provided as a Source Data file.

further analyses. INTERVAL-Old protein–PC3 components were multiplied by −1 so that the components were scaled in the same direction in all cohorts.

**Association of 90 protein biomarkers with cognitive ability.** Twenty-two proteins and protein–PC1 were associated with general fluid cognitive ability in the LBC1936 (N = 798, age ~73 years) (β between −0.11 and −0.17, p < 0.0029) (Supplementary Data 2 and Table 3). In all instances lower protein levels were associated with higher cognitive ability. Of these 23 associations, two [carboxypeptidase M (CPM) and sialic acid binding Ig like lectin 1 (SIGLEC1)] were nominally associated with general fluid cognitive ability in the age-matched INTERVAL-Old cohort (N = 975, age ≥ 67 years) (β = −0.07 and −0.08 respectively, p < 0.05). Sixteen associations were significant in a meta-analysis of the LBC1936 and INTERVAL-Old groups (β between −0.07 and −0.10, p < 0.0029) (Supplementary Data 2). The remaining seven associations were nominally significant in the meta-analysis (β between −0.05 and −0.07, p < 0.05) (Supplementary Data 2). Direction of the effect was

consistent in both cohorts, but effect sizes were smaller in INTERVAL-Old, for all 22 proteins and for protein–PC1. Fourteen of the 23 associations showed evidence of heterogeneity in the meta-analysis (ChiSq between 4.1 and 8.9, p < 0.05), indicating that the effect sizes were significantly different between the two cohorts. The protein with the strongest association, in both the LBC1936 and the meta-analysis, was ectodysplasin A2 receptor (EDA2R). When we additionally corrected cognitive ability and proteins for smoking status and antihypertensive medication use, the majority of associations were slightly attenuated, but remained significant (Supplementary Data 3). Poliovirus receptor (PVR) became the protein with the strongest association in LBC1936, and discoidin domain receptor family, member 1 (DDR1) was most strongly associated in the meta-analysis.

In the older and smaller LBC1921 (N = 165, age ~87 years), eight of the 23 proteins/protein–PC1 (including EDA2R) were nominally significantly associated with general fluid cognitive ability (β between −0.16 and −0.20, p < 0.05), and the direction of the effect was the same for all 23. The effect sizes were similar to the LBC1936 results for most of them (Supplementary Data 2).

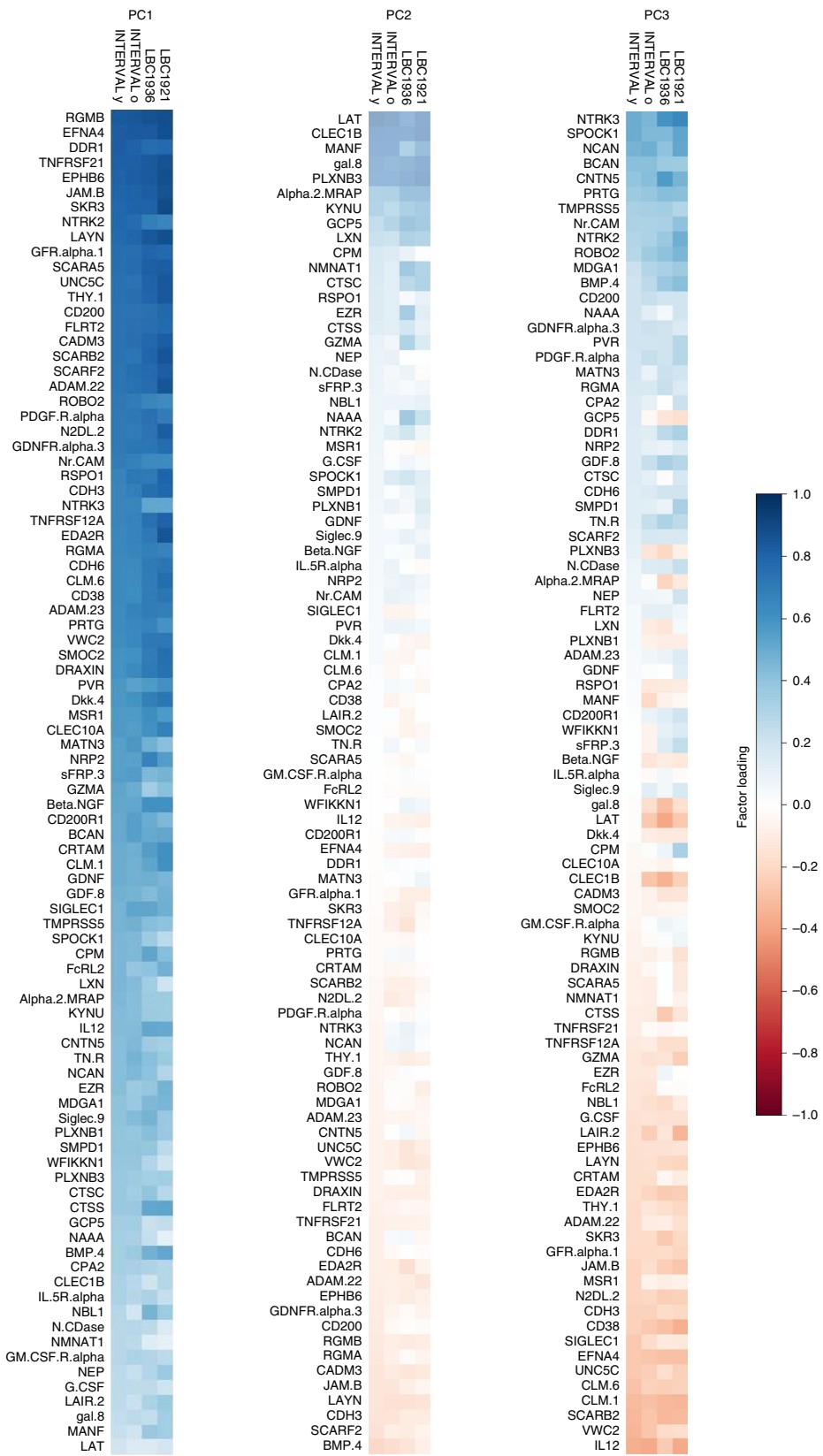

**Fig. 2 Heatmaps illustrating the loadings for individual proteins for the first three protein principal components for each cohort.** Heatmap illustrating the loadings for individual proteins for protein–PC1, PC2 and PC3 for INTERVAL-Young, INTERVAL-Old, LBC1936 and LBC1921.

**Table 3 Proteins and principal components (PCs) associated with general fluid cognitive ability in LBC1936 at $p < 0.0029$.**

| Protein | Beta | SE | P value |
|---|---|---|---|
| EDA2R | −0.17 | 0.035 | $7.32 \times 10^{-7}$ |
| PVR | −0.17 | 0.035 | $1.44 \times 10^{-6}$ |
| EFNA4 | −0.16 | 0.035 | $1.03 \times 10^{-5}$ |
| DDR1 | −0.15 | 0.035 | $3.43 \times 10^{-5}$ |
| RSPO1 | −0.14 | 0.035 | $5.70 \times 10^{-5}$ |
| SKR3 | −0.14 | 0.035 | $6.49 \times 10^{-5}$ |
| TNFRSF12A | −0.14 | 0.035 | $8.77 \times 10^{-5}$ |
| VWC2 | −0.14 | 0.035 | $1.02 \times 10^{-4}$ |
| Siglec-9 | −0.14 | 0.035 | $1.10 \times 10^{-4}$ |
| SCARB2 | −0.13 | 0.035 | $1.86 \times 10^{-4}$ |
| LAYN | −0.13 | 0.035 | $2.03 \times 10^{-4}$ |
| UNC5C | −0.13 | 0.035 | $2.77 \times 10^{-4}$ |
| CLM-6 | −0.13 | 0.035 | $2.82 \times 10^{-4}$ |
| CPM | −0.13 | 0.035 | $2.90 \times 10^{-4}$ |
| MSR1 | −0.13 | 0.035 | $3.08 \times 10^{-4}$ |
| Protein–PC1 | −0.12 | 0.035 | $4.46 \times 10^{-4}$ |
| N2DL-2 | −0.12 | 0.035 | $5.90 \times 10^{-4}$ |
| GFR-alpha-1 | −0.12 | 0.035 | $6.49 \times 10^{-4}$ |
| SIGLEC1 | −0.12 | 0.035 | $9.52 \times 10^{-4}$ |
| CDH6 | −0.12 | 0.035 | $9.80 \times 10^{-4}$ |
| THY1 | −0.11 | 0.035 | 0.0012 |
| SCARA5 | −0.11 | 0.035 | 0.0024 |
| PLXNB1 | −0.11 | 0.035 | 0.0029 |

Corrected for age and sex.

In the larger INTERVAL-Young ($N = 3476$, age $\leq 66$ years), there was no replication ($p > 0.05$) of the LBC1936 associations and direction of effect was consistent for half (12/23) of LBC1936 associations (Supplementary Data 2). Similar results were found when additionally correcting for smoking status and antihypertensive drug use (Supplementary Data 3). Supplementary Fig. 1 shows scattergraphs indicating effect sizes for 90 proteins and protein–PC1–PC3 for all four cohorts. The LBC1936 is positively correlated with INTERVAL-Old, LBC1921 and INTERVAL-Young (Pearson correlation coefficients = 0.21, 0.38 and 0.41 respectively, $p < 0.05$), indicating that, in general, associations with general cognitive function across all 90 proteins were similar in effect between the LBC1936 and the other three cohorts. A negative correlation was identified between the LBC1921 and INTERVAL-Young cohorts (Pearson correlation coefficient = −0.21, $p < 0.05$).

**Association of 90 protein biomarkers with brain variables.** Ten, seven and six proteins plus protein–PC3 were associated with total brain, grey matter and normal-appearing white matter volumes, respectively, after Bonferroni correction ($p < 0.0029$) in the LBC1936. Protein–PC3, neurocan (NCAN) and contactin 5 (CNTN5) were associated with gFA ($\beta$ between 0.14 and 0.18, $p < 0.0029$). Secreted frizzled-related protein 3 (SFRP-3), CNTN5 and cadherin 6 (CDH6) were associated with gMD ($\beta$ between −0.12 and −0.13, $p < 0.0029$) (Supplementary Data 4). No proteins or protein–PCs were associated with WMH or PVS score (all $p > 0.0029$). Twenty-two proteins and protein–PC1 were associated with general cognitive function in LBC1936; some of these were also associated with brain volume (total brain [5], grey matter [4] and normal-appearing white matter [2]), and gMD [1] ($p < 0.0029$). Similar results were found when additionally correcting for smoking status and antihypertensive drug use (Supplementary Data 5).

Higher levels of EDA2R were associated with smaller total brain volume ($\beta = -0.21$, $p = 3.9 \times 10^{-7}$), smaller grey matter volume ($\beta = -0.16$, $p = 7.4 \times 10^{-5}$) and less normal-appearing

white matter volume ($\beta = -0.2$, $p = 1.8 \times 10^{-4}$). The strongest associations with total brain, grey matter and normal-appearing white matter volumes were with NCAN and brevican (BCAN). Higher levels of NCAN and BCAN were associated with larger brain volumes ($\beta$ between 0.16 and 0.28); higher levels were also associated with higher fluid cognitive ability in INTERVAL-Young ($\beta = 0.07$, $p = 2.0 \times 10^{-5}$; $\beta = 0.06$, $p = 4.0 \times 10^{-4}$). Protein–PC3 was the only principal component associated with brain volumes ($\beta$ between 0.15 and 0.23); it was also associated with fluid cognitive ability in INTERVAL-Young ($\beta = 0.07$, $p = 6.8 \times 10^{-5}$).

**Mediation analysis in LBC1936.** Mediation analyses were performed in the LBC1936 to investigate if brain MRI phenotypes mediated the association between the 23 proteins/protein–PC1 and general fluid cognitive ability. Total brain volume corrected for intracranial volume significantly and partially mediated the association between ten of these proteins and general fluid cognitive ability (FDR-corrected, percentage attenuation between 16.2% and 35.9%) (Table 4). The most significant mediation was identified for EDA2R, where the association between higher EDA2R and poorer cognitive ability was partially (30.6%; $\beta$ reduced from −0.157 to −0.109) mediated via total brain volume (Fig. 3a). Multiple brain MRI measures mediated the association between half (5/10) of the proteins and general fluid cognitive ability (FDR-corrected, percentage attenuation between 22.0% and 36.4%) (Table 5). The most significant mediation was identified for EDA2R, where the association between higher EDA2R and poorer cognitive ability was partially (36.42%; $\beta$ reduced from −0.162 to −0.103) mediated via brain variables (Fig. 3b). Similar results were found when additionally correcting for smoking status and antihypertensive drug use (Supplementary Data 7 and 8). Figure 4 and Supplementary Data 6 show that the greatest unique contributions to this mediation effect were consistently from normal-appearing white matter and grey matter volumes.

For those proteins for which grey matter volume was a significant mediator of protein–cognitive associations (EDA2R, PVR, SKR3, MSR1 and GFR-alpha-1), we conducted a post hoc analysis of the regional distribution of protein–cortical associations. The results of the magnitude, distribution and FDR-corrected significance of these associations are shown in Fig. 5. Except for GFR-alpha-1, for which no significant associations were found, higher levels of all proteins were associated with lower cortical volumes in parts of the cingulate, lateral frontal and both anterior and medial temporal cortices. By contrast, parietal and occipital areas were markedly spared. When we additionally corrected the cortical volumes and proteins for smoking status and antihypertensive medication use, all associations were attenuated to non-significance for SKR3 (mean attenuation = 16.17%, SD = 7.63 and max = 41.73%). The attenuation found for both EDA2R (M = 10.40%, SD = 5.20 and max = 28.79%) and MSR1 (M = 10%, SD = 5.16 and max = 32.33%) was comparable, and the least attenuation was seen for associations between cortical volume and PVR (M = 3.96%, SD = 2.18 and max = 10.33%). Whereas the FDR-corrected extent of the associations was reduced in all cases, some fronto-temporal associations were still evident for EDA2R, SKR3 and PVR. Pearson correlations between normalised protein expression levels for these five proteins are shown in Supplementary Table 5. All the protein levels were moderately correlated (Pearson correlations 0.4–0.8).

**Discussion**

This study investigated associations between 90 neurology-related proteins and general fluid cognitive ability in the LBC1936,

**Table 4 Mediation of association between protein–PC1 and proteins and general fluid cognitive ability by total brain volume in LBC1936.**

| Protein | Total beta | Total SE | Total P | Total brain vol IDE beta | Total brain vol IDE SE | Total brain vol IDE P | % attn | c' |
|---|---|---|---|---|---|---|---|---|
| EDA2R | −0.157 | 0.04 | <0.001 | −0.048 | 0.012 | **<0.001** | 30.57 | −0.109 |
| PVR | −0.173 | 0.04 | <0.001 | −0.028 | 0.011 | **0.009** | 16.18 | −0.145 |
| EFNA4 | −0.135 | 0.041 | 0.001 | −0.027 | 0.011 | **0.015** | 20.00 | −0.108 |
| DDR1 | −0.138 | 0.04 | 0.001 | −0.01 | 0.01 | 0.35 | 7.25 | −0.128 |
| RSPO1 | −0.141 | 0.041 | 0.001 | −0.02 | 0.011 | 0.057 | 14.18 | −0.121 |
| SKR3 | −0.126 | 0.041 | 0.002 | −0.036 | 0.012 | **0.002** | 28.57 | −0.090 |
| TNFRSF12A | −0.114 | 0.04 | 0.005 | −0.027 | 0.011 | **0.014** | 23.68 | −0.087 |
| VWC2 | −0.124 | 0.04 | 0.002 | −0.01 | 0.01 | 0.354 | 8.06 | −0.114 |
| Siglec-9 | −0.143 | 0.04 | <0.001 | −0.014 | 0.01 | 0.184 | 9.79 | −0.129 |
| SCARB2 | −0.131 | 0.041 | 0.002 | −0.033 | 0.011 | **0.004** | 25.19 | −0.098 |
| LAYN | −0.121 | 0.041 | 0.003 | −0.026 | 0.011 | **0.019** | 21.49 | −0.095 |
| UNC5C | −0.101 | 0.04 | 0.012 | −0.015 | 0.011 | 0.147 | 14.85 | −0.086 |
| CLM-6 | −0.12 | 0.04 | 0.003 | −0.026 | 0.011 | 0.055 | 21.67 | −0.094 |
| CPM | −0.116 | 0.04 | 0.004 | −0.006 | 0.01 | 0.56 | 5.17 | −0.110 |
| MSR1 | −0.106 | 0.039 | 0.007 | −0.038 | 0.011 | **0.001** | 35.85 | −0.068 |
| Protein–PC1 | −0.113 | 0.04 | 0.005 | −0.01 | 0.01 | 0.334 | 8.85 | −0.103 |
| N2DL-2 | −0.11 | 0.04 | 0.006 | −0.019 | 0.011 | 0.079 | 17.27 | −0.091 |
| GFR-alpha-1 | −0.106 | 0.041 | 0.009 | −0.031 | 0.011 | **0.006** | 29.25 | −0.075 |
| SIGLEC1 | −0.12 | 0.04 | 0.003 | −0.028 | 0.011 | **0.011** | 23.33 | −0.092 |
| CDH6 | −0.091 | 0.04 | 0.023 | −0.006 | 0.01 | 0.572 | 6.59 | −0.085 |
| THY1 | −0.105 | 0.041 | 0.01 | −0.017 | 0.011 | 0.118 | 16.19 | −0.088 |
| SCARA5 | −0.096 | 0.041 | 0.018 | −0.004 | 0.011 | 0.722 | 4.17 | −0.092 |
| PLXNB1 | −0.079 | 0.041 | 0.052 | −0.012 | 0.011 | 0.277 | 15.19 | −0.067 |

Corrected for age and sex. Total effect sizes (total betas) differ from those in Table 3 as the mediation analysis included fewer individuals. Significant mediations (FDR corrected) are indicated in bold. *IDE* indirect effect, *% attn* percentage attenuated.

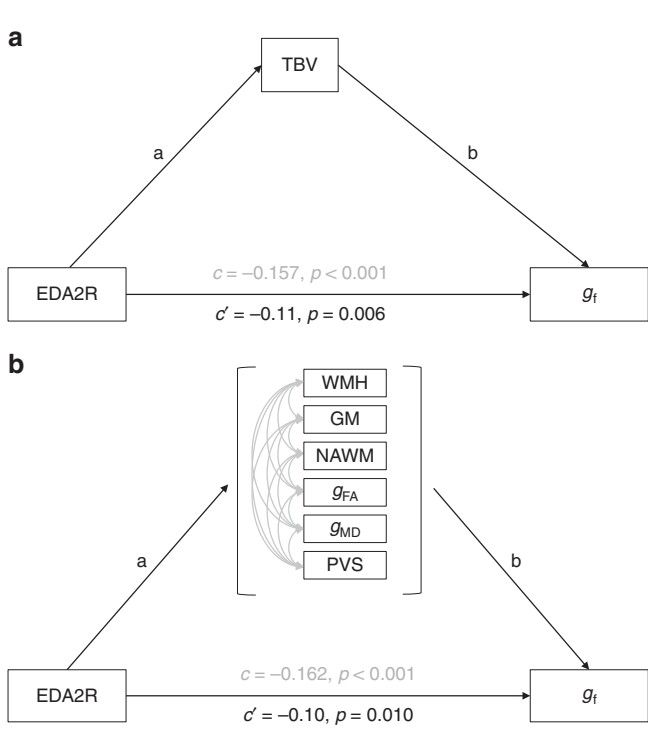

**Fig. 3 Mediation analysis.** The association between EDA2R and gf was **a** significantly partially mediated (31%) by total brain volume in LBC1936. Indirect effect (a × b) = −0.048, p < 0.001. **b** Significantly partially mediated (36%) by brain MRI variables in LBC1936. Indirect effect (a × b) = −0.059, p < 0.001.

**LBC1921 and INTERVAL.** Twenty-two proteins were associated with general fluid cognitive ability in the LBC1936 and in a meta-analysis of LBC1936 and an age-matched INTERVAL sample. Effect sizes, although smaller in INTERVAL-Old, were all in the same direction as in the LBC1936. Another study that measured proteins in two different populations, by using an Olink panel, showed similar differences in the effect sizes of associations with

insulin resistance[23]. Differences in effect sizes may be due to blood from the two cohorts being collected in different tube types (citrate for LBC1936, EDTA for INTERVAL-Old) or differences in the selection bias between the two cohorts. Similar effect sizes to LBC1936 were found for the majority of these 22 proteins in the older LBC1921, indicating that associations do not change between the ages of 73 and 87 years. No replication was identified in INTERVAL-Young, suggesting that age-related changes in protein associations with general cognitive ability may occur. Mediation analysis showed that brain volume mediated the association between ten of the proteins and general fluid cognitive ability. The two proteins that showed the strongest association with total brain, grey matter and normal-appearing white matter volumes (NCAN, BCAN) were not significantly associated with general fluid cognitive ability in the LBC1936, LBC1921 or INTERVAL-Old groups, but were associated in the INTERVAL-Young sample. Similar effect sizes for the associations with cognitive ability were found in LBC1936 and INTERVAL-Young, but these associations were not significant in the smaller LBC1936. The EDA2R protein showed the strongest association with general fluid cognitive ability in the meta-analysis of the LBC1936 and age-matched INTERVAL-Old samples. EDA2R (Ectodysplasin A2 Receptor) is a member of the type III transmembrane protein of the TNFR (tumor necrosis factor receptor) superfamily encoded by *EDA2R* on chromosome X. This protein is important in hair and tooth development[24], and levels of EDA2R have been shown to increase with age in blood[25] and lung tissue[26]. It was also associated with reactive astrogliosis in mice[27] and enriched in mouse astrocytes[28], indicating that higher levels of this protein may reduce cognitive ability by reducing the number of healthy neurons. Other proteins that were relatively strongly associated with general fluid cognitive ability in the LBC1936 and the meta-analysis of the LBC1936 and INTERVAL-Old sample included sialoadhesin encoded by the *SIGLEC1* gene on chromosome 20, a member of the immunoglobulin family[29], which may influence cognitive ability through its roles in demyelination and neuroinflammation[30]; poliovirus receptor encoded by the *PVR* gene on chromosome 19—viral infections have been previously linked to neurodegeneration[31]; R-spondin-1 encoded by the *RSPO1* gene

**Table 5 Mediation of association between protein–PC1 and proteins and general cognitive fluid ability by MRI brain variables in LBC1936: grey matter volume, normal-appearing white matter volume, white matter hyperintensity volume, perivascular spaces, general fractional anisotropy and general mean diffusivity.**

| Protein | Total beta | Total SE | Total P | Sum IDE beta | Sum IDE SE | Sum IDE P | % attn | c′ |
|---|---|---|---|---|---|---|---|---|
| EDA2R | −0.162 | 0.041 | <0.001 | −0.059 | 0.015 | **<0.001** | 36.42 | −0.103 |
| PVR | −0.173 | 0.041 | <0.001 | −0.038 | 0.014 | **0.006** | 21.97 | −0.135 |
| EFNA4 | −0.136 | 0.042 | 0.001 | −0.028 | 0.014 | 0.049 | 20.59 | −0.108 |
| DDR1 | −0.136 | 0.041 | 0.001 | −0.014 | 0.014 | 0.308 | 10.29 | −0.122 |
| RSPO1 | −0.136 | 0.041 | 0.001 | −0.027 | 0.014 | 0.055 | 19.85 | −0.109 |
| SKR3 | −0.125 | 0.042 | 0.003 | −0.038 | 0.015 | **0.01** | 30.40 | −0.087 |
| TNFRSF12A | −0.117 | 0.041 | 0.005 | −0.032 | 0.015 | 0.027 | 27.35 | −0.085 |
| VWC2 | −0.129 | 0.041 | 0.001 | −0.012 | 0.013 | 0.342 | 9.30 | −0.117 |
| Siglec-9 | −0.141 | 0.041 | 0.001 | −0.018 | 0.014 | 0.187 | 12.77 | −0.123 |
| SCARB2 | −0.136 | 0.042 | 0.001 | −0.024 | 0.015 | 0.104 | 17.65 | −0.112 |
| LAYN | −0.126 | 0.042 | 0.003 | −0.025 | 0.014 | 0.074 | 19.84 | −0.101 |
| UNC5C | −0.101 | 0.041 | 0.014 | −0.01 | 0.014 | 0.438 | 9.90 | −0.091 |
| CLM-6 | −0.118 | 0.041 | 0.004 | −0.031 | 0.015 | 0.033 | 26.27 | −0.087 |
| CPM | −0.109 | 0.041 | 0.008 | −0.014 | 0.013 | 0.299 | 12.84 | −0.095 |
| MSR1 | −0.109 | 0.04 | 0.006 | −0.039 | 0.015 | **0.007** | 35.78 | −0.070 |
| Protein–PC1 | −0.111 | 0.041 | 0.007 | −0.006 | 0.013 | 0.646 | 5.41 | −0.105 |
| N2DL-2 | −0.109 | 0.041 | 0.008 | −0.018 | 0.014 | 0.218 | 16.51 | −0.091 |
| GFR-alpha-1 | −0.107 | 0.042 | 0.01 | −0.036 | 0.014 | **0.01** | 33.64 | −0.071 |
| SIGLEC1 | −0.112 | 0.041 | 0.007 | −0.021 | 0.014 | 0.12 | 18.75 | −0.091 |
| CDH6 | −0.088 | 0.041 | 0.032 | 0.002 | 0.014 | 0.862 | −2.27 | −0.090 |
| THY1 | −0.104 | 0.041 | 0.012 | −0.021 | 0.014 | 0.121 | 20.19 | −0.083 |
| SCARA5 | −0.098 | 0.041 | 0.018 | −0.005 | 0.014 | 0.718 | 5.10 | −0.093 |
| PLXNB1 | −0.081 | 0.041 | 0.050 | −0.006 | 0.014 | 0.636 | 7.41 | −0.075 |

Corrected for age and sex. Total effect sizes (total betas) differ from those in Table 3 as the mediation analysis included fewer individuals. Significant mediations (FDR corrected) are indicated in bold. *IDE* indirect effect, *% attn* percentage attenuated.

on chromosome 1 and expressed in the central nervous system during development[32]; discoidin domain receptor family, member 1 encoded by the *DDR1* gene on chromosome 6, which is important in myelination[33]. The addition of smoking status and antihypertensive drug use as covariates slightly attenuated many of the results.

Interestingly, two chondroitin sulfate proteoglycans (CSPGs) that are common constituents of the extracellular matrix (ECM) and specific to the CNS were strongly associated with brain volume in LBC1936. CSPGs are key members of perineuronal nets (PNNs), which are ECM structures surrounding neurons, important in storage and maintenance of long-term memories[34–39]. Neurocan and brevican are encoded by *NCAN* (chromosome 19) and *BCAN* (chromosome 1), respectively, and are expressed in astrocytes and neurons. BCAN is also expressed in oligodendrocytes. These were the only CSPGs on the Olink assay. Neurocan inhibits neuronal adhesion and neurite outgrowth in vitro[40]. Common genetic variation in NCAN is associated with bipolar disorder[41]. NCAN is the closest relative of BCAN, and animal knockouts of BCAN and NCAN have a similar phenotype (normal development and memory with deficient hippocampal long-term potentiation)[42,43]. NCAN peaks in development and declines in the adult brain. In contrast, BCAN is one of the most common CSPGs in the adult brain. It is not yet known what role CSPGs and the PNN may play in age-related cognitive decline; however, our data suggest that NCAN and BCAN are associated with brain volume and may potentially play a neuroprotective role for general fluid cognitive ability in early adulthood. Although expression of NCAN and BCAN is highly specific to the brain, we have shown that levels detected in plasma, in which it is much easier to obtain samples of, also correlate with brain structure. Future studies will be required to confirm these proteins as blood biomarkers of brain structure.

PCA indicated that the levels of the individual proteins were not independent, with 30% of the variance explained by the first

PC. The first three PCs derived from the 90 proteins were highly congruent between the four cohorts, providing cross-sample validation of the stability of the proteins' correlational structure. The first PC was associated with general fluid cognitive ability in the LBC1936, LBC1921, and a meta-analysis of LBC1936 and INTERVAL-Old samples. This association was not mediated by brain variables in the LBC1936, suggesting that the influence on general fluid cognitive ability was independent of the micro- and macrostructural brain variables measured at the global level. Proteins that loaded highly on protein–PC1 included RGM domain family member B (RGMB) that is involved in patterning of the developing nervous system[44], and Ephrin-A4 (EFNA4) and Ephrin type-B receptor 6 (EPHB6), both of which are members of the ephrin family that is implicated in the development of the nervous system[45]. Our data suggest that these proteins may also be important in the ageing nervous system. These findings can serve to sharpen downstream mechanistic and molecular work on the role of specific proteins in processes involved in CNS ageing. Protein–PC3 (like BCAN and NCAN that load highly on protein–PC3) was not associated with general fluid cognitive ability, but was associated with total brain, grey matter and normal-appearing white matter volumes in the LBC1936, suggesting that although it is related to brain volume, it does not do so in a way that affects general fluid cognitive ability. A review looking at how components of PNNs, including BCAN and NCAN, control plasticity, and on their role in memory in normal ageing, concluded that interventions that target PNNs may allow the brain to function well, despite pathology[36]. Therefore, components of the PNN may protect against changes in brain volume.

The fairly common pattern of protein–cortical associations in the cingulate, temporal and frontal lobes is of interest, as these are among the regions implicated in higher cognitive function[46–49]. The five proteins in these analyses showed a moderate level of correlation, but despite this the same vascular risk-type covariates (smoking and hypertension) lead to slightly different levels of

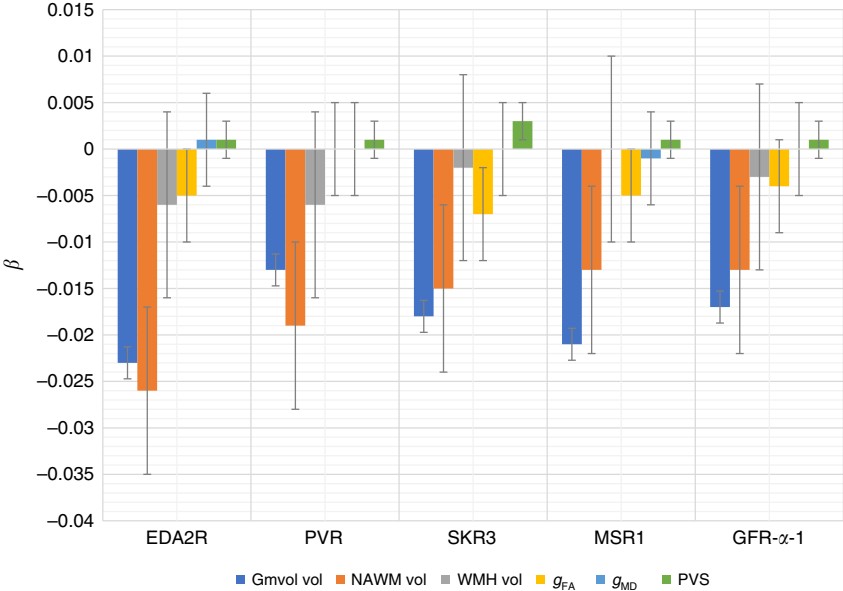

**Fig. 4 Mediation analysis in LBC1936 separated by brain variable.** Unique contributions from each of the brain variables are indicated with standard error bars. Gmvol = grey matter volume, NAWM vol = normal-appearing white matter volume, WMH vol = white matter hyperintensity volume, gFA = general fractional anisotropy, gMD = general mean diffusivity, PVS = perivascular spaces.

attenuation. As was shown in the analyses looking at protein levels and general cognitive ability the least attenuation was identified for PVR. The fact that these vascular risk factors attenuated the associations might indicate the differential relevance of these specific blood biomarkers in the well-established associations between vascular risk and brain structure[50].

The strengths of this study include the fact that protein levels, cognitive ability and structural brain variables were measured in the same individuals at about the same time in ~600 members of the LBC1936. Participants in the LBC1936 have a narrow age range and are an ancestrally homogeneous population, which reduces the variability compared with other cohorts. The age-matched INTERVAL cohort for replication of associations with general fluid cognitive ability and the ability to investigate these associations in both an older (LBC1921) and younger (INTER-VAL-Young) cohort were further strengths of this study, giving a total sample size larger than most other studies of this type. A key strength of the INTERVAL sample is that they are all healthy blood donors, which minimises confounding by disease status. The Olink Neurology panel was particularly well suited to this study as all proteins were chosen because of a prior link to neurology-related diseases, traits or processes and because it has high sensitivity and specificity[20].

The limitations of the study included the fact that the proteins were measured in blood rather than brain tissue. However, as blood samples are relatively easy to obtain, proteins in the blood that are associated with cognitive function and brain structure are more likely to be useful as future biomarkers. Also, a panel of pre-selected neurology-related proteins was used, rather than bespoke assays for proteins that we specifically hypothesised to be associated with cognitive ability and brain structure. One other potential limitation of our investigation is the use of non-fasting plasma samples. However, a recent study concluded that timing of food intake only had a modest effect on the levels of the Olink neurology-related biomarkers used in this study[51]. The use of citrate blood collection tubes for the LBCs and EDTA blood collection tubes for INTERVAL is potentially a limitation. However, the fact that the within-protein correlational structure

was consistent across cohorts, suggests that it was not a significant confound. Another limitation was the lack of a replication cohort that included brain MRI variables. A further limitation is that we investigated cognitive measures at the global level. Potentially counterintuitive findings (such as the protein–PC3 associations with brain volumes but not general fluid cognitive ability) are plausible where specific cognitive abilities are affected. A further potential limitation is the use of different cognitive tests in the LBC1936, the LBC1921 and the INTERVAL sample. Although research has shown that general factors created from different cognitive batteries are highly consistent[52,53], and specifically in LBC1936 two general cognitive function phenotypes calculated from two non-overlapping batteries of cognitive tests had a correlation of $r = 0.79$[9], a more ideal study would have administered the same cognitive tests to each cohort and extracted a general factor from the combined cohorts.

In conclusion, we have identified several proteins associated with general fluid cognitive ability and brain volume that should be replicated in an independent study before being considered as reliable and possibly useful biomarkers of cognitive ability in later life. Integrating information about these proteins with information about established biomarkers for dementia, such as amyloid β42 and neurofilament light, may help to identify biological pathways to potentially target therapeutically for age-related cognitive decline.

## Methods

**Lothian Birth Cohort 1936.** LBC1936 consists of 1091 individuals, most of whom took part in the Scottish Mental Survey of 1947 at the age of ~11 years old. In the survey, they took a validated test of cognitive ability, the Moray House Test (MHT) version 12[54]. They were recruited to a study to determine influences on cognitive ageing at age ~70 years and have taken part in four waves of testing in later life (at mean ages 70, 73, 76 and 79 years). At each wave they underwent a series of cognitive and physical tests, with concomitant brain MRI introduced at age ~73 years[21]. For this study, cognitive tests were performed, and plasma was extracted from blood collected in citrate tubes at a mean age of 72.5 (SD 0.7) years. The cognitive tests included here were six of the non-verbal subtests from the Wechsler Adult Intelligence Scale-IIIUK (WAIS-III)[55]: matrix reasoning, letter-number sequencing, block design, symbol search, digit symbol coding and digit span backwards. From these six cognitive tests, a general fluid cognitive component was

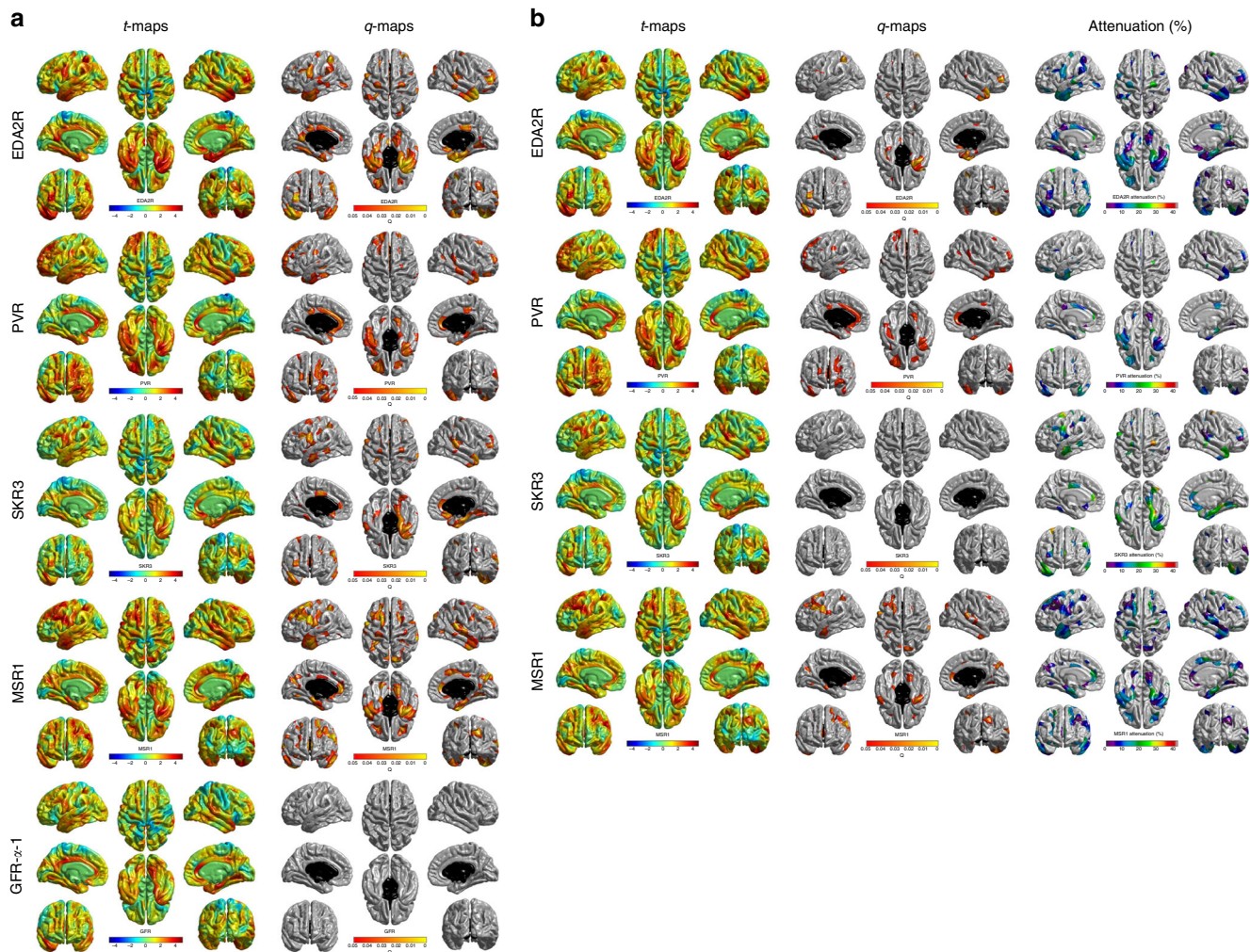

**Fig. 5 Regional distribution of protein–cortical associations. a** Corrected for age and sex, **b** corrected for age, sex, smoking status and antihypertensive medication use and the percentage attenuation due to the addition of smoking status and antihypertensive medication use. Source data are provided as a Source Data file.

derived. The scores from the first unrotated component of a principal component analysis were extracted and labelled as general fluid cognitive ability. This component explained 51% of the variance, with individual test loadings ranging from 0.65 to 0.76. General fluid cognitive ability was regressed onto age and sex (and separately onto age, sex, smoking status and antihypertensive drug use), and residuals from these linear regression models were used in further statistical analyses. Cognitive data and neurology-related protein levels were available for 798 individuals. In all, 7% of these individuals self-reported stroke, 0.2% dementia and 0.4% Parkinson's disease. No other neurological conditions were reported.

Whole-brain structural and diffusion tensor MRI data were acquired by using a 1.5 T GE Signa Horizon scanner (General Electric, Milwaukee, WI, USA) located at the Brain Research Imaging Centre, University of Edinburgh, soon after cognitive testing and plasma collection. Mean age at scanning was 72.7 (SD 0.7) years. Full details are given in ref. [56]. In brief, T1-, T2-, T2* and FLAIR-weighted MRI sequences were collected and co-registered (voxel size = $1 \times 1 \times 2$ mm). Total brain, grey matter, normal-appearing white matter volume and WMH were calculated by using a semi-automated multispectral fusion method[16,57,58]. PVS were visually rated (5-point score in basal ganglia and centrum semiovale; the sum of the two scores was used in this study) by a trained neuroradiologist[16].

The diffusion tensor MRI protocol employed a single-shot spin-echo echo-planar diffusion-weighted sequence in which diffusion-weighted volumes ($b = 1000$ s mm$^{-2}$) were acquired in 64 non-collinear directions, together with seven T$_2$-weighted volumes ($b = 0$ s mm$^{-2}$). This protocol was run with 72 contiguous axial slices with a field of view of $256 \times 256$ mm, an acquisition matrix of $128 \times 128$ and 2-mm isotropic voxels. Full details are included in ref. [16].

White matter connectivity data were created by using the BEDPOSTX/ ProbTrackX algorithm in FSL (https://fsl.fmrib.ox.ac.uk), and 12 major tracts of interest were segmented using Tractor (https://www.tractor-mri.org.uk) scripts: the genu and splenium of the corpus callosum, and bilateral anterior thalamic radiations, cingulum bundles, uncinate, arcuate and inferior longitudinal fasciculi.

Tract-average white matter FA and MD were derived as the average of all voxels contained within the resultant tract maps. General factors of FA (gFA) and MD (gMD) were derived from a confirmatory factor analysis using all 12 tracts, to reflect the well-replicated phenomenon of common microstructural properties of brain white matter in early, middle and later life[59–61]. Each of the T1-weighted volumes were processed using FreeSurfer v5.1. Following visual quality control in which the outputs for each participant were inspected for aberrant surface meshes, skull stripping and tissue segmentation failures, their estimated cortical surfaces were registered to the 'fsaverage' template, yielding a measure of regional volume at each of 327,684 vertices across the cortical mantle.

WMH volume was log transformed, after which it showed an approximately normal distribution. Total brain, grey matter, normal-appearing white matter volume and log WMH volumes were regressed onto age, sex and intracranial volume (and separately onto age, sex, intracranial volume, smoking status and antihypertensive drug use). PVS score, gFA and gMD were regressed onto age and sex (and separately onto age, sex, smoking status and antihypertensive drug use). Residuals from these linear regression models were used in further statistical analyses. Brain imaging data and neurology-related plasma protein levels were available for between 600 and 635 individuals.

**Lothian Birth Cohort 1921.** LBC1921 consists of 550 individuals, most of whom took part in the Scottish Mental Survey of 1932 at the age of ~11 years old. In the survey, they took a validated test of cognitive ability, the MHT version 12[62]. They were recruited to a study to determine influences on cognitive ageing at age ~79 years and have taken part in five waves of testing in later life (at ages 79, 83, 87, 90 and 92 years). For this study, cognitive tests were performed, and plasma was extracted from blood collected in citrate tubes at a mean age of 86.6 years (SD 0.4)[21]. Cognitive tests included Raven's Standard Progressive Matrices[63], letter-number sequencing[55] and digit symbol coding[55]. From these three cognitive tests, a general

fluid cognitive component was derived. The scores from the first unrotated component of a principal component analysis were extracted and labelled as general fluid cognitive ability. This component explained 68% of the variance, with individual test loadings ranging from 0.78 to 0.83. General fluid cognitive ability was regressed onto age and sex, and residuals from these linear regression models were used in further statistical analyses. Cognitive data and neurology-related plasma protein levels were available for 165 individuals. In all, 8% of these individuals self-reported stroke, 0.6% dementia and 0% Parkinson's disease. No other neurological conditions were reported.

**INTERVAL**. INTERVAL is a randomised trial of ~45,000 blood donors from the National Health Service Blood and Transplant Centres in England[22]. The trial was designed to determine whether the interval between donations could be safely reduced. Cognitive function tests were taken ~2 years into the trial at which point plasma was extracted from blood collected in EDTA tubes. Cognitive tests adapted from the Cardiff Cognitive Battery[64] were assessed: Stroop Test (part 1, measures attention and reaction times in milliseconds); Trail Making Test (duration of part B in milliseconds, measures executive function); Pairs Test (participants were asked to memorise the positions of six card pairs, and then match them from memory while making as few errors as possible) and Reasoning Test (a task with 13 logic/reasoning-type questions and a 2-min time limit). Scores on the Pairs Test are for the number of errors that each participant made; higher scores reflect poorer episodic memory. The Reasoning Test is known as the 'Fluid Intelligence' test in UK Biobank[10]. The scores from the first unrotated component of a PCA of the four tests were extracted and labelled as general fluid cognitive ability. This component explained 48% of the variance, with individual test loadings ranging from 0.35 to 0.60. General fluid cognitive ability was regressed onto age and sex (and separately onto age, sex, smoking status, antihypertensive drug use and trial arm), and residuals from these linear regression models were used in further statistical analyses. Cognitive data and neurology-related protein biomarkers were available for 4451 individuals. INTERVAL is a relatively cognitively healthy cohort, as history of stroke, Alzheimer's disease, dementia, Parkinson's disease or other neurological conditions make an individual ineligible to donate blood; see https://my.blood.co.uk/KnowledgeBase/. For the purposes of this study, INTERVAL was split into individuals aged ≥ 67 (INTERVAL-Old, N = 975, mean age = 70.3 years, SD = 2.4 years), and individuals ≤ 66 years (INTERVAL-Young, N = 3476, mean age = 58.4 years, SD = 5.0 years). The former subsample was formed to be approximately matched in mean age with the LBC1936.

**Neurology-related protein biomarker measurement**. In total, 92 neurology-related protein biomarkers were measured in plasma by the Proximity Extension Assay technique by using the Proseek Multiplex Neurology I 96 × 96 reagents kit by Olink® Proteomics[19]. Storage times for all plasma samples within each cohort were similar. The data were pre-processed by Olink® using NPX Manager software. Supplementary Table 6 lists the percentage of samples below Olink's pre-determined lower limit of detection (LLOD) for LBC and INTERVAL, for each protein. Proteins with more than 10% of samples below the LLOD were removed from further analyses. For the remaining proteins all values including those below the LLOD were included. Normalised protein expression levels were transformed by inverse-rank normalisation, to avoid potential false positives caused by outlying values and then regressed onto age and sex (and separately onto age, sex, smoking status and antihypertensive drug use [plus trial arm allocation in INTERVAL]). Residuals from these linear regression models were used in further statistical analyses. LBC1936 and LBC1921 used a newer version of the kit, which included microtubule-associated protein tau (MAPT) rather than brain-derived neurotrophic factor (BDNF). Both BDNF (in INTERVAL) and MAPT (in the LBCs) failed quality control, as did HAGH in both cohorts, and were therefore excluded from all analyses. See Supplementary Table 1 for the 90 proteins analysed.

**Statistical analyses**. We conducted a PCA of the 90 proteins for each cohort to establish the common variance among these markers. We used the coefficient of factor congruence to assess the consistency with which the individual proteins loaded on each component across groups. We used PCA results to inform our threshold for multiple testing of independent tests (number of components with eigenvalues >1). PCA on the transformed levels of the 90 neurological markers revealed that 17 components explained the majority (70%) of the variance in the data in the LBC1936. Based on PCA, a Bonferroni-corrected p value of 0.0029 (0.05/17 independent proteins) was used to indicate statistical significance[65].

Next, linear regression models were used to test the associations of each of the 90 neurology-related protein biomarkers with general fluid cognitive ability (LBC1936, LBC1921 and INTERVAL-Old and Young), total brain, grey matter, normal-appearing white matter and WMH volumes, and PVS, gFA and gMD (LBC1936 only). We also extracted the first three components from the PCA of all 90 proteins that showed acceptable stability across cohorts, i.e. those with a coefficient of factor congruence > 0.70. We then examined their associations with cognitive and brain variables, as above. Linear regression analyses were performed in R[66]. The results from LBC1936 and the approximately age-matched INTERVAL-Old cohort were inverse variance weighted fixed-effect meta-analysed using (METAL)[67].

Finally, we performed mediation analysis in a structural equation modelling framework to identify if the significant (Bonferroni-corrected) protein–cognitive ability associations were mediated by the brain MRI variables in the LBC1936. Two analyses were performed. The first included total brain volume corrected for intracranial volume. The second included multiple brain structural mediators (grey matter, normal-appearing white matter and WMH volumes, all corrected for intracranial volume), PVS, gFA and gMD. For these analyses no selection for brain imaging variables was made on the basis of their association with the proteins. Mediation analyses were carried out by using the lavaan package, using bootstrapping to calculate the standard errors, in R[66].

Brain cortical volumetric analyses were conducted using the SurfStat toolbox (http://www.math.mcgill.ca/keith/surfstat) for Matrix Laboratory R2018a (The MathWorks Inc., Natick, MA), for which 595 participants had complete MRI, protein and covariate data.

**Reporting summary**. Further information on research design is available in the Nature Research Reporting Summary linked to this article.

## Data availability

LBC data supporting the findings of this paper are available from the corresponding author upon reasonable request. The INTERVAL Study Group has previously published its trial protocol, statistical analysis plan, informed consent form and other relevant study documents. Bona fide scientists can seek access to relevant de-identified individual participant data (and a copy of the trial's data dictionary) by applying to the INTERVAL Data Access Committee after print publication of this paper at the following e-mail address: helpdesk@intervalstudy.org.uk. The INTERVAL Data Access Committee reviews (supplemented, when required, by expertise from additional scientists external to the committee) applications according to usual academic criteria of scientific validity and feasibility. Following approval by the INTERVAL Data Access Committee, a material transfer or research collaboration agreement will be agreed and signed with the applicants. Applicants might be requested to provide reimbursement of data management or preparation costs, as the INTERVAL trial is no longer in receipt of funding. Applicants will be required to provide updates to the INTERVAL Data Access Committee on their use of the INTERVAL trial data, including provision of copies of any publications. Applicants will be required to adhere in publications with the INTERVAL trial's policy for acknowledgement of the trial's funders, stakeholders and scientific or technical contributors. The source data underlying Figs. 1a and 5 are provided as a Source Data file.

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

## Acknowledgements

*Lothian Birth Cohorts 1921 and 1936:* We thank the cohort participants and team members who contributed to these studies. Phenotype collection in the Lothian Birth Cohort 1921 was supported by the United Kingdom's Biotechnology and Biological Sciences Research Council (BBSRC), The Royal Society and The Chief Scientist Office of the Scottish Government. Phenotype collection in the Lothian Birth Cohort 1936 was supported by Age UK (The Disconnected Mind project). LBC1936 MRI brain imaging was supported by Medical Research Council (MRC) grants G0701120, G1001245, MR/M013111/1 and MR/R024065/1. The Olink® Proteomics assays were supported by a National Institutes of Health (NIH) research grant R01AG054628. The work was undertaken by The University of Edinburgh Centre for Cognitive Ageing and Cognitive Epidemiology, a part of the cross- council Lifelong Health and Wellbeing Initiative;

funding from the BBSRC and MRC is gratefully acknowledged (MR/K026992/1). SRC, MEB, IJD and EMTD were supported by NIH research grant R01AG054628. MVH was supported by the Row Fogo Charitable Trust (Grant No. BROD.FID3668413). We gratefully acknowledge the contribution of co-author Professor John M. Starr, who died prior to the publication of this paper.

*INTERVAL:* Participants in the INTERVAL randomised controlled trial were recruited with the active collaboration of NHS Blood and Transplant England (www.nhsbt.nhs.uk), which has supported field work and other elements of the trial. DNA extraction and genotyping was co-funded by the National Institute for Health Research (NIHR), the NIHR BioResource (http://bioresource.nihr.ac.uk/) and the NIHR [Cambridge Biomedical Research Centre at the Cambridge University Hospitals NHS Foundation Trust] [*]. The Olink® Proteomics assays were funded by Biogen, Inc. (Cambridge, MA, USA). The academic coordinating centre for INTERVAL was supported by core funding from NIHR Blood and Transplant Research Unit in Donor Health and Genomics (NIHR BTRU-2014-10024), UK Medical Research Council (MR/L003120/1), British Heart Foundation (SP/09/002; RG/13/13/30194; RG/18/13/33946) and the NIHR [Cambridge Biomedical Research Centre at the Cambridge University Hospitals NHS Foundation Trust] [*]. A complete list of the investigators and contributors to the INTERVAL trial is provided in reference [**]. The academic coordinating centre would like to thank blood donor centre staff and blood donors for participating in the INTERVAL trial.

*The views expressed are those of the authors and not necessarily those of the NHS, the NIHR or the Department of Health and Social Care.

**Di Angelantonio E, Thompson SG, Kaptoge SK, Moore C, Walker M, Armitage J, Ouwehand WH, Roberts DJ, Danesh J, INTERVAL Trial Group. Efficiency and safety of varying the frequency of whole blood donation (INTERVAL): a randomised trial of 45,000 donors. Lancet. 2017 Nov 25;390(10110):2360–2371.

## Author contributions

S.E.H., S.R.C., S.B., R.E.M. and I.J.D. participated in the design and/or interpretation of the reported experiments or results. S.E.H., S.R.C., S.B., R.E.M., B.P.P., A.P., J.C., S.M.M., M.V.H., Z.M., S.J., P.G.B., E.M.T.D., J.M.S., M.E.B., J.M.W., A.S.B. and I.J.D. participated in the acquisition and/or analysis of data. S.E.H., S.R.C., S.B., R.E.M., B.P.P., A.P., J.C., S.M.M., M.V.H., Z.M., S.J., P.G.B., E.M.T.D., M.E.B., J.M.W., A.S.B. and I.J.D. participated in drafting and/or revising the paper.

## Competing interests

The authors declare no competing interests.

## Ethical approval

Ethics permission for the LBC1936 was obtained from the Scotland A Research Ethics Committee (07/MRE00/58). Ethics permission for the LBC1921 was obtained from the Lothian Research Ethics Committee (1702/98/4/183). The INTERVAL trial has received ethics committee approval from the National Research Ethics Service Committee East of England- (REC 11/EE/0538). All persons gave their informed consent prior to their inclusion in the study.
