## [Peer Review File · Nature Communications]

Reviewers' comments:

Reviewer #1 (Remarks to the Author):

This is an interesting investigation of the associations between a targeted proteomics panel (by an interesting technology, OLINK, which created a panel of proteins previously associated with a variety of neurological disorders) assayed in blood from three large cohorts and metrics of general fluid cognitive ability; moreover, the study assesses the possible mediation of the association by global MRI measures (total brain volume, total GM, WMH, and similar). The study attempts to take advantage of "big data" to address interesting questions. Unfortunately, this choice results in heterogeneity in methods, which in my view is not adequately addressed. I remain unconvinced that the study has revealed some valid and important signatures rather than noise, but I would like to give the authors the opportunity to address these concerns.

- There is huge heterogeneity in the tests used to create the general fluid cognitive ability among the different cohorts, which is recognized as a limitation in the end of the discussion, but in a superficial way. It is unclear whether the combined metric represents the same cognitive domain or set of domains. Are there any cohorts, in which some common tests were employed that may allow us to look at cross-correlations? Any literature support for this particular construct? It is unclear how metric that combines "matrix reasoning, letter-number sequencing, block design, symbol search, digit symbol coding, and digit span backwards", in one cohort, may be compared with a metric that combines "Raven's Standard Progressive Matrices, letter-number sequencing and digit symbol coding" in another.
- Similarly, it is unclear to what extent PC1, 2, 3 in one cohort is comparable to PC1, PC2, PC3 in the other cohorts. The authors treat them as more or less identical entities. I sorted the top loadings of individual proteins in the xl spreadsheet provided and noticed some common top loadings, but also substantial differences. Why was PCA conducted separately for each cohort? If it was conducted in a combined super cohort, the PCs would be actually the same. No convincing explanation is provided.
- Some obvious conceptual limitations are not recognized or discussed: the presumed relationship of blood proteins levels with their brain expression, the fact that the proteins were included in the panel semi-arbitrarily, some of them may not even be brain expressed or enriched.
- A VBM type study looking at regional associations may have been more revealing of neural correlates than whole brain measures. Why were the latter selected

Reviewer #2 (Remarks to the Author):

Harris et al conducted a study of circulating proteins in peripheral blood in three population-based samples. The study materials seem relevant and sufficiently large for meaningful statistical analyses. The MRI measures and cognitive tests have been conducted to high standards, although no longitudinal data on cognitive function are presented. Also, the Olink PEA method is by now well-established for multiplex measurement of circulating proteins. While circulating serum proteins seem less relevant for brain-related phenotypes than cerebrospinal fluid proteins, there are certainly proteins that correlate well between CSF and serum such as Neurofilament light, making the overall analysis still relevant.

There is certainly a big need to understand the biology implicated in cognitive function and combining cognitive tests with carefully measured circulating proteins should be of interest in the field.

The presentation of the work by Harris could be improved in some aspects. In particular, I lack a clear hypothesis statements with background on what was previously known about protein

biomarkers in peripheral blood in relation to cognitive function and MRI brain measures. The introduction and presentation of results are technically written but don't emphasize the results that were consistent across study cohorts. I'm uncertain if the authors see the INTERVAL cohort as true replication since results are not described as such.

I have some concerns with the statistical analyses of Olink protein data. It is well known that proteins in serum or plasma are influenced by numerous other factors, most importantly age, gender, sample handling, freezer time, medications, smoking etc. The manuscript states that "Normalised protein expression levels were transformed by inverse-rank normalisation and then regressed onto age and sex" but I can't seem to find how the authors managed the variance introduced by factors other than age and gender in their statistical analysis. Furthermore, and importantly, the Olink analysis paragraph should explain how values below Olink's pre-determined Lower Limit of Detection (LLOD) was handled i.e. not all proteins will be detectable in all samples.

The discussion would benefit from some additional thoughts on how the current results may yield insights into cognitive decline and dementias, and especially in relation to established biomarkers in the field such as serum amyloid-beta or Neurofilament light. It's possible that I missed information on whether any of the participants in the study cohorts had developed disorders related to cognitive function e.g. MCI, AD, PD etc. If such clinical events were present, were those subjects included in the present analyses?

Minor comments:

Lines 74-78. Please state whether the analyses are cross-sectional, prospective or both.

Line 86: change biomarker to protein

Line 103: Should the reader consider the 2 replicated associations as the key significant association with cognitive ability or the 16 associations in the meta-analysis?

Line 107: Were the protein levels significantly different between the different cohorts analysed?

Line 110: Do the authors by "most significant" mean that it was the protein with the strongest association with cognitive ability? If so, please rephrase as the current wording is confusing.

Line 113: What is meant by "nominally significantly associated"? I was under the impression that the significance level had been set with a Bonferroni correction according to the number of PCs explaining 70 % of the total variance?

Line 132: Here it states "Bonferroni correction" but what was it Bonferroni corrected for: the number of proteins and traits analysed (i.e. 91x3) or something else?

Line 187: It is stated that "The two proteins that showed the strongest association with total brain, grey matter and normal appearing white matter volumes (NCAN, BCAN) were not associated with general fluid cognitive ability in the LBC1936, LBC1921 or INTERVAL-Old groups, but were associated in the INTERVAL-Young sample." but the authors don't speculate on potential explanations for the cohort differences.

Line 191: Text says "EDA2R is the tumour necrosis factor receptor superfamily member 27" but I guess what's meant is that EDA2R codes for "protein name".

Line 203: The NCAN/BCAN discussion seems relevant but would the authors care to speculate on how brain NCAN/BCAN are related to serum/plasma levels?

Line 214: The conclusion that the PCA indicated that proteins were not independent seems simplistic to me. Several of the proteins are strongly correlated with age and gender, and I'm

suspecting that the PCA results reflect the population structure rather than the pair-wise correlation between proteins at a biological level.

Reviewer #3 (Remarks to the Author):

In this study, Harris and colleagues performed targeted proteomic analyses on several very large aging cohorts to examine relationships between neurology-related proteins, fluid cognitive abilities, and structural and functional brain imaging measures. They used retrospectively collected plasma specimens from approximately 5,400 individuals and compared expression of 91 plasma proteins with a general fluid cognitive ability score based on cognitive tests and several imaging measures (e.g., FA, total brain volume, white matter hyperintensity volume, etc). Using a principle components analysis approach, they found that roughly 25% of the measured proteins (22/91) were associated with the general fluid cognitive ability score and that total brain volume mediated this effect for 10/22 proteins. They conclude that a number of proteins are associated with general fluid cognitive ability and brain volume and that these may be useful as biomarkers of cognitive ability in later life and to identify biological pathways to target for age-related cognitive decline.

This is a paper from a prominent and well-respected research group. The overwhelming strength of the study is the very large cohort of well characterized participants which provides the potential to discover biomarkers that are truly generalizable at the population level. However, large sample size here does not entirely mitigate some significant concerns with the methodology and overall treatment of the findings.

First, there is no clear hypothesis for the study. The authors state only that they want to explore relationships between protein expression, cognition, and brain imaging in older adults. The rationale for the 92 protein platform is not clear; only that the proteins are "neurology-related". It would have been preferred if the proteins had been selected a priori in order to investigate their role in the outcomes.

There are several methodological concerns regarding the proteomic data. It's not clear when the plasma samples were collected relative to the proteomic analysis. Specifically, they need to give the length of time the samples were in the freezer before analysis. If the 1936 birth cohort was on average 72.5 years old at time of sample collection, then these samples were collected in 2008 and may have been in the freezer for nearly a decade before proteomic analysis. Proteins degrade with time and this is now an expected piece of information when reporting fluid biomarker studies. Because storage time affects protein expression it would be preferred to use storage time as a covariate in the analysis.

The authors note that prandial state was not controlled for, but medications in this population could have an even greater effect. Medications or classes of medications were not reported.

The use of citrate tubes for two LBC studies and EDTA tubes for INTERVAL is problematic. Although mentioned as a limitation, this is a real problem that could affect protein expression and may be reason why some findings were not replicated across the cohorts.

General fluid cognitive ability was measured using different tests and the authors state that this is a limitation to the study. This is always a challenge in large studies like these. However, it's surprising that they did not attempt to deconstruct the general cognitive measure to look at relationships between individual tests and protein abundance; especially since they raise this point in the Discussion.

The discussion is cursory and only a few specific proteins in PCA 1 were discussed. In the end,

they really didn't attempt to provide biological meaning for the majority of the proteins they found loading onto PCA1.

Figure 4 has no ordinate label and there is no indication what the units are. In addition, the errors bars are not indicated.

Reviewer #1

This is an interesting investigation of the associations between a targeted proteomics panel (by an interesting technology, OLINK, which created a panel of proteins previously associated with a variety of neurological disorders) assayed in blood from three large cohorts and metrics of general fluid cognitive ability; moreover, the study assesses the possible mediation of the association by global MRI measures (total brain volume, total GM, WMH, and similar). The study attempts to take advantage of "big data" to address interesting questions. Unfortunately, this choice results in heterogeneity in methods, which in my view is not adequately addressed. I remain unconvinced that the study has revealed some valid and important signatures rather than noise, but I would like to give the authors the opportunity to address these concerns.

Response: We are pleased that the reviewer finds our investigation interesting, and hope that our responses below convince them that the study has revealed some valid and important signatures, by explicitly testing the suggested sources of noise.

- There is huge heterogeneity in the tests used to create the general fluid cognitive ability among the different cohorts, which is recognized as a limitation in the end of the discussion, but in a superficial way. It is unclear whether the combined metric represents the same cognitive domain or set of domains. Are there any cohorts, in which some common tests were employed that may allow us to look at cross-correlations? Any literature support for this particular construct? It is unclear how metric that combines "matrix reasoning, letter-number sequencing, block design, symbol search, digit symbol coding, and digit span backwards", in one cohort, may be compared with a metric that combines "Raven's Standard Progressive Matrices, letter-number sequencing and digit symbol coding" in another.

Response: There is indeed literature supporting the use of this particular construct using different batteries of cognitive tests. As cited in the original manuscript, Johnson et al., 2004 and 2008 showed that general factors created from different cognitive batteries are highly consistent (all correlations $r > 0.77$). This high correlation was also shown specifically in the LBC1936 cohort, whereby two general fluid-type cognitive function component phenotypes were derived, each using a different battery of cognitive tests. The first battery consisted of six non-verbal tests from the Wechsler Adult Intelligence Scale-III UK; these were Block Design, Digit Symbol, Symbol Search, Letter-Number Sequencing, Backward Digit Span, and Matrix Reasoning. The second battery contained the Moray House Test, Logical Memory, Spatial Span, Four Choice Reaction Time, and Verbal Fluency. These two general cognitive function phenotypes, calculated from two non-overlapping batteries of cognitive tests had a correlation of $r = 0.79$ ($P < 0.001$) (Davies et al 2015). The following text has been added to the discussion section on page 18:

"and specifically in LBC1936 two general cognitive function phenotypes calculated from two non-overlapping batteries of cognitive tests had a correlation of $r = 0.79$ ($P < 0.001$)⁹"

- Similarly, it is unclear to what extent PC1, 2, 3 in one cohort is comparable to PC1, PC2, PC3 in the other cohorts. The authors treat them as more or less identical entities. I sorted the top loadings of individual proteins in the xl spreadsheet provided and noticed some common top loadings, but also substantial differences. Why was PCA conducted separately for each cohort? If it was conducted in a combined super cohort, the PCs would be actually the same. No convincing explanation is provided.

Response: The coefficient of factor congruence between the four cohorts ranged between |0.70 to 1.00| for the first three principal components indicating that they are highly comparable between cohorts. This is the formal psychometric method for comparing the comparability of principal

components. We have added the relevant heatmaps to the supplementary information Fig. S1; these indicate the loadings for individual proteins, which illustrates the similarity of the PCs between cohorts. Due to restrictions on the raw data, it is not possible to conduct PCA in a combined super cohort, as LBC and INTERVAL raw data are stored and analysed in different institutions. In any case, it is not necessary, because, as we now show more clearly, the PCs are highly comparable between cohorts.

- Some obvious conceptual limitations are not recognized or discussed: the presumed relationship of blood proteins levels with their brain expression, the fact that the proteins were included in the panel semi-arbitrarily, some of them may not even be brain expressed or enriched.

Response: We have added the following text to the limitations section of the discussion, page 17:

“Limitations of the study included the fact that the proteins were measured in blood rather than brain tissue. However, as blood samples are relatively easy to obtain, proteins in the blood that are associated with cognitive function and brain structure are more likely to be useful as future biomarkers. Also, a panel of pre-selected neurology-related proteins was used, rather than bespoke assays for proteins that we specifically hypothesised to be associated with cognitive ability and brain structure.”

- A VBM type study looking at regional associations may have been more revealing of neural correlates than whole brain measures. Why were the latter selected

Response: We thank the reviewer for this suggestion, which we have implemented. We have now used the results from the well-characterised global measures of brain micro- and macrostructure to guide a more detailed analysis, and have now performed brain cortical volume analyses looking at specific regions of grey matter. The following text has been added to the manuscript:

Methods page 20:

“Each of the T1-weighted volumes were processed using FreeSurfer v5.1. Following visual quality control in which the outputs for each participant were inspected for aberrant surface meshes, skull stripping and tissue segmentation failures, their estimated cortical surfaces were registered to the ‘fsaverage’ template, yielding a measure of regional volume at each of 327,684 vertices across the cortical mantle.”

Methods page 23:

“Brain cortical volumetric analyses were conducted using the SurfStat toolbox (<http://www.math.mcgill.ca/keith/surfstat>) for Matrix Laboratory R2018a (The MathWorks Inc., Natick, MA), for which 595 participants had complete MRI, protein and covariate data.”

Results page 12:

“For those proteins for which grey matter volume was a significant mediator of protein-cognitive associations (EDA2R, PVR, SKR3, MSR1, GFR-alpha-1), we conducted a post-hoc analysis of the regional distribution of protein-cortical associations. Results of the magnitude, distribution and FDR-corrected significance of these associations are shown in Fig S3. Except for GFR-alpha-1, for which no significant associations were found, higher levels of all proteins were associated with lower cortical volumes in parts of the cingulate, lateral frontal and both anterior and medial temporal cortices. By contrast, parietal and occipital areas were markedly spared. When we additionally corrected the cortical volumes and proteins for smoking status and antihypertensive medication use, all associations were attenuated to non-significance for SKR3 (Mean attenuation =

16.17 %, SD = 7.63, max = 41.73%). The attenuation found for both EDA2R (M = 10.40 %, SD = 5.20, max = 28.79%) and MSR1 (M = 10.00 %, SD = 5.16, max = 32.33%) were comparable, and the least attenuation was seen for associations between cortical volume and PVR (M = 3.96 %, SD = 2.18, max = 10.33%). Whereas the FDR-corrected extent of the associations were reduced in all cases, some fronto-temporal associations were still evident for EDA2R, SKR3 and PVR.”

Discussion page 16:

“The fairly common pattern of protein-cortical associations in the cingulate, temporal and frontal lobes is of interest, as these are among the regions implicated in higher cognitive function⁴²⁻⁴⁵. The five proteins in these analyses showed a moderate level of correlation, but despite this the same vascular risk-type covariates (smoking and hypertension) lead to slightly different levels of attenuation. As was shown in the analyses looking at protein levels and general cognitive ability, the least attenuation was identified for PVR. The fact that these vascular risk factors attenuated the associations might indicate the differential relevance of these specific blood biomarkers in the well-established associations between vascular risk and brain structure⁴⁶”

Reviewer #2

Harris et al conducted a study of circulating proteins in peripheral blood in three population-based samples. The study materials seem relevant and sufficiently large for meaningful statistical analyses. The MRI measures and cognitive tests have been conducted to high standards, although no longitudinal data on cognitive function are presented. Also, the Olink PEA method is by now well-established for multiplex measurement of circulating proteins. While circulating serum proteins seem less relevant for brain-related phenotypes than cerebrospinal fluid proteins, there are certainly proteins that correlate well between CSF and serum such as Neurofilament light, making the overall analysis still relevant.

Response: We thank the reviewer for these positive comments.

There is certainly a big need to understand the biology implicated in cognitive function and combining cognitive tests with carefully measured circulating proteins should be of interest in the field.

The presentation of the work by Harris could be improved in some aspects. In particular, I lack a clear hypothesis statements with background on what was previously known about protein biomarkers in peripheral blood in relation to cognitive function and MRI brain measures. The introduction and presentation of results are technically written but don't emphasize the results that were consistent across study cohorts. I'm uncertain if the authors see the INTERVAL cohort as true replication since results are not described as such.

Response: The LBC1936 was the only cohort with MRI brain variables measured. Therefore, INTERVAL-Old was seen as a replication sample for the cognitive ability analyses only. LBC1921 and INTERVAL-Young were included to examine if the cognitive ability associations were consistent in other age groups; this is what is termed 'constructive replication' (Lykken, 1968). We hope that this is clearer in the revised manuscript, which now includes the following text on page 5 of the introduction:

“...~170 members of the older Lothian Birth Cohort 1921 (LBC1921)²¹ and ~4,500 members of INTERVAL, split into a LBC196 age-matched subsample and a younger subsample to investigate if associations were consistent across different age groups.”

We have added the following hypothesis statement to the end of the introduction, page 5:

“We hypothesised that some of the neurology-related proteins would be associated with general fluid cognitive ability in older individuals, and that some of these associations would be mediated by structural brain variables.”

and the following sentence earlier in the introduction to indicate previous work looking at blood proteins and cognitive ability and brain variables, page 4:

“Peripheral blood proteins including inflammatory markers^{14,15} and S100B¹⁶ have previously been associated with cognitive ability and/or MRI brain measures...”

The following has been added to the discussion, page 14:

“Similar effect sizes to LBC1936 were found for the majority of these 22 proteins in the older LBC1921, indicating that associations do not change between the ages of 73 and 87 years. No replication was identified in INTERVAL-Young suggesting that age-related changes in protein associations with general cognitive ability may occur.”

I have some concerns with the statistical analyses of Olink protein data. It is well known that proteins in serum or plasma are influenced by numerous other factors, most importantly age, gender, sample handling, freezer time, medications, smoking etc. The manuscript states that “Normalised protein expression levels were transformed by inverse-rank normalisation and then regressed onto age and sex” but I can’t seem to find how the authors managed the variance introduced by factors other than age and gender in their statistical analysis.

Response: We are grateful for this comment and we have now taken into account the factors suggested by the referee. We have repeated the analyses adjusting for smoking status and antihypertension medication use (in addition to age and sex which were included in the original models), in the three largest cohorts (LBC1936, INTERVAL-Old and INTERVAL-Young), where these data were available. These results are presented in Supplementary Tables 7, 9, 11 and 12, and described in the text. Most effect sizes were slightly attenuated, but overall the results are very similar. Within each cohort, samples were handled and frozen for similar time periods and so no corrections were made for these.

Furthermore, and importantly, the Olink analysis paragraph should explain how values below Olink’s pre-determined Lower Limit of Detection (LLOD) was handled i.e. not all proteins will be detectable in all samples.

Response: We apologise for omitting this important information from the methods. We used all the data including values below the LLOD. We inverse-rank normalised the data to avoid potential false positives caused by outlying values. The following text has been added to the methods, page 22:

“ All values including those below Olink’s pre-determined Lower Limit of Detection were included. Normalised protein expression levels were transformed by inverse-rank normalisation, to avoid potential false positives caused by outlying values.”

The discussion would benefit from some additional thoughts on how the current results may yield insights into cognitive decline and dementias, and especially in relation to established biomarkers in the field such as serum amyloid-beta or Neurofilament light.

Response: We have added the following text to the discussion, page 18:

“Integrating information about these proteins with information about established biomarkers for dementia, such as amyloid 642 and neurofilament light, may help to identify biological pathways to potentially target therapeutically for age-related cognitive decline.”

It’s possible that I missed information on whether any of the participants in the study cohorts had developed disorders related to cognitive function e.g. MCI, AD, PD etc. If such clinical events were present, were those subjects included in the present analyses?

Response: No participants were removed from the analyses based on their developing disorders related to cognitive function. These data are not well ascertained, but the cohorts used consist of relatively healthy individuals and as such, numbers of individuals with these disorders will be very small.

Minor comments:

Lines 74-78. Please state whether the analyses are cross-sectional, prospective or both.

Response: We have now clarified that the analyses were cross-sectional.

Line 86: change biomarker to protein

Response: We have changed biomarker to protein.

Line 103: Should the reader consider the 2 replicated associations as the key significant association with cognitive ability or the 16 associations in the meta-analysis?

Response: We have presented the results as they are, and suggest that any association be replicated in an independent study, before being considered as a biomarker for cognitive ability in later life. We have added the following text to the discussion, page 18:

“In conclusion we have identified a number of proteins associated with general fluid cognitive ability and brain volume that should be replicated in an independent study before being considered as reliable and possibly useful biomarkers of cognitive ability in later life.”

Line 107: Were the protein levels significantly different between the different cohorts analysed?

Response: Due to the data normalization and standardization process used by Olink prior to their sending us the protein expression data, it is not possible to directly compare protein levels between the LBC1936 and INTERVAL-Old.

Line 110: Do the authors by “most significant” mean that it was the protein with the strongest association with cognitive ability? If so, please rephrase as the current wording is confusing.

Response: The wording has been changed to:

“The protein with the strongest significant association, in both the LBC1936 and the meta-analysis, was ectodysplasin A2 receptor (EDA2R).”

Line 113: What is meant by “nominally significantly associated”? I was under the impression that the significance level had been set with a Bonferroni correction according to the number of PCs explaining 70 % of the total variance?

Response: By nominally significantly associated we mean that the p-value was <0.05. We judged that it was important to indicate these associations in LBC1921 because the majority of the effect sizes were similar to those in LBC1936, but due to the smaller sample size they did not reach Bonferroni corrected significance.

Line 132: Here it states “Bonferroni correction” but what was it Bonferroni corrected for: the number of proteins and traits analysed (i.e. 91x3) or something else?

Response: As stated in the statistical analyses section on page 22: “PCA on the transformed levels of the 91 neurological markers revealed that 17 components explained the majority (70%) of the variance in the data in the LBC1936. Based on PCA, a Bonferroni corrected p value of 0.0029 (0.05/17 independent proteins) was used to indicate statistical significance.”

Line 187: It is stated that “The two proteins that showed the strongest association with total brain, grey matter and normal appearing white matter volumes (NCAN, BCAN) were not associated with general fluid cognitive ability in the LBC1936, LBC1921 or INTERVAL-Old groups, but were associated in the INTERVAL-Young sample.” but the authors don't speculate on potential explanations for the cohort differences.

Response: We suggest that this was possibly a power issue with LBC1936. Similar effect sizes for the associations with cognitive ability were actually found in LBC1936 and INTERVAL-Young, but these associations were not significant in the smaller LBC1936. We have added the following text to the discussion, page 14:

“Similar effect sizes for the associations with cognitive ability were found in LBC1936 and INTERVAL-Young, but these associations were not significant in the smaller LBC1936.”

Line 191: Text says “EDA2R is the tumour necrosis factor receptor superfamily member 27” but I guess what's meant is that EDA2R codes for “protein name”.

Response: EDA2R is the protein symbol for the tumour necrosis factor receptor superfamily member 27. The wording has been changed to:

EDA2R (Ectodysplasin A2 Receptor) is a member of the type III transmembrane protein of the TNFR (tumor necrosis factor receptor) superfamily encoded by *EDA2R* on chromosome X.

Line 203: The NCAN/BCAN discussion seems relevant but would the authors care to speculate on how brain NCAN/BCAN are related to serum/plasma levels?

Response: We have added the following sentence to the discussion, page 15:

“Although expression of NCAN and BCAN is highly specific to the brain, we have shown that levels detected in plasma, which it is much easier to obtain samples of, also correlate with brain structure. Future studies will be required to confirm these proteins as blood biomarkers of brain structure.”

Line 214: The conclusion that the PCA indicated that proteins were not independent seems simplistic to me. Several of the proteins are strongly correlated with age and gender, and I'm suspecting that the PCA results reflect the population structure rather than the pair-wise correlation between proteins at a biological level.

Response: We have repeated the PCA correcting for age and sex and the percentage of the variance explained by the first PC remains close to 30% for each cohort: LBC1936=33.8%; LBC1921=32.1%; INTERVAL Young=28.7%; INTERVAL Old=29.0%. These results suggest that the original PCA results do not reflect the population structure, but are due to pair-wise correlation between proteins.

Reviewer #3 (Remarks to the Author):

In this study, Harris and colleagues performed targeted proteomic analyses on several very large aging cohorts to examine relationships between neurology-related proteins, fluid cognitive abilities, and structural and functional brain imaging measures. They used retrospectively collected plasma specimens from approximately 5,400 individuals and compared expression of 91 plasma proteins with a general fluid cognitive ability score based on cognitive tests and several imaging measures (e.g., FA, total brain volume, white matter hyperintensity volume, etc). Using a principle components analysis approach, they found that roughly 25% of the measured proteins (22/91) were associated with the general fluid cognitive ability score and that total brain volume mediated this effect for 10/22 proteins. They conclude that a number of proteins are associated with general fluid cognitive ability and brain volume and that these may be useful as biomarkers of cognitive ability in later life and to identify biological pathways to target for age-related cognitive decline.

This is a paper from a prominent and well-respected research group. The overwhelming strength of the study is the very large cohort of well characterized participants which provides the potential to discover biomarkers that are truly generalizable at the population level. However, large sample size here does not entirely mitigate some significant concerns with the methodology and overall treatment of the findings.

First, there is no clear hypothesis for the study. The authors state only that they want to explore relationships between protein expression, cognition, and brain imaging in older adults. The rationale for the 92 protein platform is not clear; only that the proteins are “neurology-related”. It would have been preferred if the proteins had been selected a priori in order to investigate their role in the outcomes.

Response: We would like to thank the reviewer for their positive evaluation of their work and hope that we are able to mitigate some of their initial concerns in this response.

We have added the following hypothesis to the end of the introduction, page 5:

“We hypothesised that some of the neurology-related proteins would be associated with general fluid cognitive ability in older individuals, and that some of these associations would be mediated by structural brain variables.”

We agree that we would have preferred it if the proteins had been selected a priori in order to investigate their role in the outcomes. However, the cost of multiple individual assays on over 5000 samples is prohibitively expensive. We have added the following text to the limitations section of the discussion, page 17:

“Also, a panel of pre-selected neurology-related proteins was used, rather than bespoke assays for proteins that we specifically hypothesised to be associated with cognitive ability and brain structure.”

There are several methodological concerns regarding the proteomic data. It’s not clear when the plasma samples were collected relative to the proteomic analysis. Specifically, they need to give the

length of time the samples were in the freezer before analysis. If the 1936 birth cohort was on average 72.5 years old at time of sample collection, then these samples were collected in 2008 and may have been in the freezer for nearly a decade before proteomic analysis. Proteins degrade with time and this is now an expected piece of information when reporting fluid biomarker studies. Because storage time affects protein expression it would be preferred to use storage time as a covariate in the analysis.

Response: While we agree that plasma storage time is important, storage times for all plasma samples within each cohort were similar.

The authors note that prandial state was not controlled for, but medications in this population could have an even greater effect. Medications or classes of medications were not reported.

Response: We have conducted new analyses adjusting for smoking status and antihypertensive medication use. These results are now presented in Supplementary Tables 7, 9, 11 and 12, and described in the text. Some effect sizes were slightly attenuated, but overall the results are very similar.

The use of citrate tubes for two LBC studies and EDTA tubes for INTERVAL is problematic. Although mentioned as a limitation, this is a real problem that could affect protein expression and may be reason why some findings were not replicated across the cohorts.

Response: We agree that this is a limitation to the study and may have contributed to lack of replication. Nevertheless, the fact that the within-protein correlational structure was consistent across cohorts, may not readily sit with the hypothesis that tube type was a significant confound. We have added the following text to the discussion, page 17:

“The use of citrate blood collection tubes for the LBCs and EDTA blood collection tubes for INTERVAL is potentially a limitation. However, the fact that the within-protein correlational structure was consistent across cohorts, suggest that it was not a significant confound.”

General fluid cognitive ability was measured using different tests and the authors state that this is a limitation to the study. This is always a challenge in large studies like these. However, it's surprising that they did not attempt to deconstruct the general cognitive measure to look at relationships between individual tests and protein abundance; especially since they raise this point in the Discussion.

Response: As different cohorts took different cognitive tests, we judged that we had greater power to investigate associations with general cognitive fluid ability, rather than with specific cognitive tests. Also, most of the individual tests have strong loadings on the general cognitive ability measure, and the general cognitive ability-related variance is the source of a substantial proportion of ageing effects on cognitive tests' scores. Thus, we used a good measure of the most likely source of cognitive variation, one that is comparable across the samples here, and also kept our type 1 error rate as low as possible.

The discussion is cursory and only a few specific proteins in PCA 1 were discussed. In the end, they really didn't attempt to provide biological meaning for the majority of the proteins they found loading onto PCA1.

Response: The following text has been added to the discussion, page 16:

“Proteins that loaded highly on protein-PC1 included: RGM domain family member B (RGMB) that is involved in patterning of the developing nervous system⁴⁰; and Ephrin-A4 (EFNA4) and Ephrin

type-B receptor 6 (EPHB6), both of which are members of the ephrin family that is implicated in the development of the nervous system⁴¹. Our data suggest that these proteins may also be important in the ageing nervous system. These findings can serve to sharpen downstream mechanistic and molecular work on the role of specific proteins in processes involved in CNS ageing.”

Figure 4 has no ordinate label and there is no indication what the units are. In addition, the errors bars are not indicated.

Response: We apologise for these omissions. The ordinate label and error bar definitions have now been added.

References

Johnson, W., Bouchard, T. J., Krueger, R. F., McGue, M. & Gottesman, I. I. Just one g: consistent results from three test batteries. *Intelligence* **32**, 95–107 (2004).

Johnson, W., te Nijenhuis, J. & Bouchard, T. J. Still just 1 g: Consistent results from five test batteries. *Intelligence* **36**, 81–95 (2008).

Davies, G., et al. Genetic contributions to variation in general cognitive function: a meta-analysis of genome-wide association studies in the CHARGE consortium (N=53 949). *Mol. Psychiatry* **20**, 183-192 (2015)

Lykken, D. T. Statistical significance in psychological research. *Psychological Bulletin* **70**(3, Pt.1), 151-159 (1968).

Reviewers' comments:

Reviewer #1 (Remarks to the Author):

The authors made a good-faith effort to address the previous criticisms and the manuscript has been substantially revised and improved upon. I remain somewhat unconvinced that the PCs derived separately represent the same cognitive features across cohorts and, similarly, regarding the fluid intelligence metrics derived from different tests. I would rather see these limitations and the cohort heterogeneity being acknowledged even more clearly and not brushed aside. Nevertheless, this is an overall worthy effort and I would like to see it published, provided that limitations are more clearly stated.

Reviewer #2 (Remarks to the Author):

Many thanks for responding to and addressing previous comments. I think the presentation of the manuscript has improved.

I have only one outstanding comment: The inclusion of Olink NPX values below LLOD makes sense as it avoids having to statistically deal with cut-off left-tails of the protein distributions, and also avoids tied values if simple imputation methods are used. However, I was unable to find data showing the % of samples that were below LOD for each of the proteins or information on whether this was considered in the QC.

I am unconcerned if values below LLOD are included for proteins with a small proportion below detectable levels but if the proportion is 20-25% or higher, a lot of noise is introduced to the analysis. Moreover, a PCA analysis will be sensitive to noisy low values and can sometimes pick up just differences in sample quality rather than the data structure of protein expression. My suggestion would be to present a table of % below LLOD for each protein and introduce a filter based on that number. Hopefully (and probably) the vast majority of proteins would be carried forward after filtering. There will of course be some work associated with removing proteins that didn't survive the filter.

Reviewer #3 (Remarks to the Author):

In this revision, Harris and colleagues have made minor changes to address some of the three reviewer's concerns. The addition of heatmaps showing consistency of the individual proteins in the top principle components is a welcome addition. It is reassuring that at least the component structure is similar across the cohorts. However, many of the remaining additions, revisions, and responses to the reviewer's concerns fall short. A fundamental hypothesis, missing in the original manuscript and requested by all Reviewers, now feels like a post-hoc justification of the work that was done, rather than evidence that the work was thoughtfully constructed in order to answer a specific question about the biology of cognitive aging. In this revised manuscript, we still do not know if any, or more likely, how many participants had diseases affecting cognition such as stroke, Alzheimer's disease, dementia, Parkinson's disease, etc. A major point raised by both Reviewers 2 and 3 concerning variables affecting protein expression was only partly addressed in this revised manuscript. The authors selected a few variables- age, sex, antihypertensive medications, and smoking history as representative which is helpful, but no post-collection variables related to handling and storage were examined. The statement in the response that "...storage times for all plasma samples within each cohort were similar." is reassuring, but this or data backing up this statement should appear in the manuscript. Finally, there still is not significant discussion of the underlying biology of the significant proteins and why they might be mechanistically related to cognitive aging.

Reviewer #1

The authors made a good-faith effort to address the previous criticisms and the manuscript has been substantially revised and improved upon. I remain somewhat unconvinced that the PCs derived separately represent the same cognitive features across cohorts and, similarly, regarding the fluid intelligence metrics derived from different tests. I would rather see these limitations and the cohort heterogeneity being acknowledged even more clearly and not brushed aside. Nevertheless, this is an overall worthy effort and I would like to see it published, provided that limitations are more clearly stated.

Response: We agree that a more ideal study would have used the same tests in each cohort and have added the following text to the limitations section of the discussion, page 19.

“Although, research has shown that general factors created from different cognitive batteries are highly consistent^{48,49} and specifically in LBC1936 two general cognitive function phenotypes calculated from two non-overlapping batteries of cognitive tests had a correlation of $r=0.79$ ⁹, a more ideal study would have administered the same cognitive tests to each cohort and extracted a general factor from the combined cohorts.”

Reviewer #2

Many thanks for responding to and addressing previous comments. I think the presentation of the manuscript has improved.

I have only one outstanding comment: The inclusion of Olink NPX values below LLOD makes sense as it avoids having to statistically deal with cut-off left-tails of the protein distributions, and also avoids tied values if simple imputation methods are used. However, I was unable to find data showing the % of samples that were below LOD for each of the proteins or information on whether this was considered in the QC.

I am unconcerned if values below LLOD are included for proteins with a small proportion below detectable levels but if the proportion is 20-25% or higher, a lot of noise is introduced to the analysis. Moreover, a PCA analysis will be sensitive to noisy low values and can sometimes pick up just differences in sample quality rather than the data structure of protein expression. My suggestion would be to present a table of % below LLOD for each protein and introduce a filter based on that number. Hopefully (and probably) the vast majority of proteins would be carried forward after filtering. There will of course be some work associated with removing proteins that didn't survive the filter.

Response: We thank the reviewer for this suggestion. We now include a table showing the percentage of samples below the LLOD for each protein (Supplementary Table 14). We have repeated all relevant analyses after removing proteins where more than 10% of samples were below the LLOD. This resulted in the loss of a single protein, HAGH, but did not materially alter our overall conclusions.

Reviewer #3

In this revision, Harris and colleagues have made minor changes to address some of the three reviewer's concerns. The addition of heatmaps showing consistency of the individual proteins in the

top principle components is a welcome addition. It is reassuring that at least the component structure is similar across the cohorts.

Response: We are pleased that the reviewer liked the addition of the heatmaps showing consistency of the individual proteins in the top principal components.

However, many of the remaining additions, revisions, and responses to the reviewer's concerns fall short. A fundamental hypothesis, missing in the original manuscript and requested by all Reviewers, now feels like a post-hoc justification of the work that was done, rather than evidence that the work was thoughtfully constructed in order to answer a specific question about the biology of cognitive aging.

Response: We are sorry that the reviewer thinks that the hypothesis feels like a post-hoc justification of the work, but we judge that testing a broad panel of neurology-related proteins, rather than specific proteins which together form a known pathway or are involved in a particular function, for their association with cognitive ability and brain structure is an economically viable way to uncover previously unknown associations between such proteins, cognitive function and brain structure. As stated on the Olink website: "The panel offers a mix of established markers related to neurobiological processes and neurological diseases (e.g. neural development, axon guidance, synaptic function, or specific conditions such as Alzheimer's disease), as well as some more exploratory proteins with broader roles in processes such as cellular regulation, immunology, development and metabolism. This balanced selection provides an ideal basis for protein biomarker discovery in the neurology area". This relatively hypothesis free method has the strength of being able to uncover "hidden" associations and nuances of association between brain structure and cognitive function (eg, proteins associated with cognitive function with no clear association with brain phenotypes, proteins that correlate with both "adverse" cognitive function and brain variables, etc) that can then be targeted for further investigation in follow-up studies.

In this revised manuscript, we still do not know if any, or more likely, how many participants had diseases affecting cognition such as stroke, Alzheimer's disease, dementia, Parkinson's disease, etc. A major point raised by both Reviewers 2 and 3.

Response: A history of stroke, Alzheimer's disease, dementia, Parkinson's disease or other neurological condition would make an individual ineligible to donate blood (see <https://my.blood.co.uk/KnowledgeBase/>), therefore, the number of individuals with these conditions in INTERVAL is expected to be negligible or non-existent. 8% of LBC1921 participants self-report stroke, 0.6% dementia and 0% Parkinson's Disease. No other neurological conditions were reported. 7% of LBC1936 participants self-report stroke, 0.2% dementia and 0.4% Parkinson's Disease. No other neurological conditions were reported. This information has been added to pages 20 and 22 of the manuscript.

Concerning variables affecting protein expression was only partly addressed in this revised manuscript. The authors selected a few variables- age, sex, antihypertensive medications, and smoking history as representative which is helpful, but no post-collection variables related to handling and storage were examined. The statement in the response that "...storage times for all plasma samples within each cohort were similar." is reassuring, but this or data backing up this statement should appear in the manuscript.

Response: The following text has been added to page 23 of the manuscript:

"Storage times for all plasma samples within each cohort were similar."

Finally, there still is not significant discussion of the underlying biology of the significant proteins and why they might be mechanistically related to cognitive aging.

Response: Further discussion of the underlying biology of the significant proteins and why they might be mechanistically related to cognitive ageing has been added to the discussion, pages 15 and 16. The relevant section with the new text underlined is shown below.

"This protein is important in hair and tooth development²⁴ and levels of EDA2R have been shown to increase with age in blood²⁵ and lung tissue²⁶. It was also associated with reactive astrogliosis in mice²⁷ and enriched in mouse astrocytes²⁸, indicating that higher levels of this protein may reduce cognitive ability by reducing the number of healthy neurons. Other proteins that were relatively strongly associated with general fluid cognitive ability in the LBC1936 and the meta-analysis of the LBC1936 and INTERVAL-Old sample included sialoadhesin encoded by the SIGLEC1 gene on chromosome 20, a member of the immunoglobulin family²⁹, which may influence cognitive ability through its roles in demyelination and neuroinflammation³⁰; poliovirus receptor encoded by the PVR gene on chromosome 19 – viral infections have been previously linked to neurodegeneration³¹; R-spondin-1 encoded by the RSPO1 gene on chromosome 1 and expressed in the central nervous system during development³²; and discoidin domain receptor family, member 1 encoded by the DDR1 gene on chromosome 6, which is important in myelination³³."

REVIEWERS' COMMENTS:

Reviewer #2 (Remarks to the Author):

The authors have adequately addressed my comment regarding analysis of Olink proteins under the limit of detection and I was pleased to see the inclusion of the table describing % samples below LOD. I don't have any outstanding comments.

Reviewer #3 (Remarks to the Author):

The fact that the protein panel was selected in order to be inclusive and not restricted to test a particular biological hypothesis necessitates some discussion of the biological relevance of the protein markers, which has now been added. This is a welcome addition precisely because of the open ended approach to the protein panel selection.